# Epigenetic landscape of pancreatic neuroendocrine tumours reveals distinct cells of origin and means of tumour progression

Annunziata Di Domenico [1,2], Christodoulos P. Pipinikas[3], Renaud S. Maire[1], Konstantin Bräutigam [1], Cedric Simillion[4], Matthias S. Dettmer [1], Erik Vassella[1], Chrissie Thirlwell[3,5], Aurel Perren [1] & Ilaria Marinoni[1✉]

Recent data suggest that Pancreatic Neuroendocrine Tumours (PanNETs) originate from α- or β-cells of the islets of Langerhans. The majority of PanNETs are non-functional and do not express cell-type specific hormones. In the current study we examine whether tumour DNA methylation (DNAme) profiling combined with genomic data is able to identify cell of origin and to reveal pathways involved in PanNET progression. We analyse genome-wide DNAme data of 125 PanNETs and sorted α- and β-cells. To confirm cell identity, we investigate ARX and PDX1 expression. Based on epigenetic similarities, PanNETs cluster in α-like, β-like and intermediate tumours. The epigenetic similarity to α-cells progressively decreases in the intermediate tumours, which present unclear differentiation. Specific transcription factor methylation and expression vary in the respective α/β-tumour groups. Depending on DNAme similarity to α/β-cells, PanNETs have different mutational spectra, stage of the disease and prognosis, indicating potential means of PanNET progression.

[1] Institute of Pathology, University of Bern, Murtenstrasse 31, 3008 Bern, Switzerland. [2] Graduate School for Cellular and Biomedical Sciences, University of Bern, 3010 Bern, Switzerland. [3] UCL Cancer Institute, 72, Huntley Street, London WC1E 6JD, UK. [4] Bioinformatics and Computational Biology, University of Bern, Baltzerstrasse 6, 3012 Bern, Switzerland. [5] University of Exeter, College of Medicine and Health, St Luke's Campus, Heavitree Road, Exeter EX1 2LU, UK. ✉email: ilaria.marinoni@pathology.unibe.ch

Pancreatic neuroendocrine tumours (PanNETs) are tumours of the islets of Langerhans. The cell of origin is unclear, and mechanisms associated with progression are largely unknown. Surgery is currently the only curative option; however, 5-year disease free survival is ~50% in patients following an R0 resection[1]. To date, there is no validated risk prediction tool to accurately guide follow-up and select patients at high risk of recurrence who might benefit from adjuvant therapy[2]. PanNETs are clinically and genetically heterogeneous; ~40% of patients present with mutations in either DAXX or ATRX and MEN1, which encode for proteins involved in epigenetic regulation[3]. PanNETs with mutations in DAXX or ATRX have a shorter disease free survival compared to wild-type tumours[4].

The islets of Langerhans include five different cell types producing specific hormones: glucagon is produced by α-cells, insulin by β-cells, somatostatin by δ-cells, ghrelin by ε-cells and pancreatic polypeptide by PP-cells. Only a minority of PanNETs are functional, secreting inadequate hormones that lead to clinical syndromes. The majority of functional PanNETs are insulinomas. Whether functional tumours and non-functional tumours originate from the same cell type remains uncertain.

Recent studies of gene expression and master regulator analysis alongside investigation of super-enhancer signatures have suggested both α- and β-cells as two possible cells of origin for non-functioning (NF)-PanNETs[5–8]. On the other hand, Sadanandam et al. reported that a group of aggressive PanNET, namely, "metastasis-like primary", have a phenotype characterised by "stemness" transcripts (in terms of pancreatic progenitor-specific genes) compared to well-differentiated tumours, also suggesting a common progenitor cell origin[5]. Similarly, based on the identification of master regulator proteins, dedifferentiation and acquisition of stem cell characteristics seem to be one of the pathways associated with tumour progression[6].

The cell of origin in cancer refers to the normal cell that acquires the initial cancer-promoting genetic hit(s). During development, cell lineage fate is determined by cell-type-specific transcription factor (TF) expression, which in turn is dependent on the type of epigenetic markers that are located at the relative regulatory regions (e.g. super-enhancer activation)[9]. The five endocrine cell types derive from a common endocrine precursor, which has segregated from a ducto-endocrine bipotent cell population[10]. The TFs, Pax4 (Paired Box 4) and Arx (Aristaless Related Homeobox), are required for β- and α-cell fates, respectively. Lineage decision is determined via cross-inhibitory interactions[11]. Pdx1 (Pancreatic And Duodenal Homeobox 1) expression becomes restricted to cells at the stage of initiating insulin expression and, in the pancreatic islets of Langerhans, remains up-regulated exclusively in β-cells[12]. Integrative analysis of human epigenomes including histone modification patterns, DNA accessibility, DNA methylation and RNA expression has revealed that disease- and trait-associated genetic variants are enriched in tissue-specific epigenomic marks[13]. In the context of tumour biology, epigenetic states of cell lineages shape the vulnerability for specific genetic alterations and thereby reveal a distinct class of lineage-associated cancer genes[14–16]. Therefore, determining the cell of origin is crucial to understand tumour specific carcinogenesis and progression[17]. Cancer DNA-methylation profiles have been utilised to determine the cell of origin of several tumour types[18,19]. In this study, we set out to determine the putative cell of origin of PanNETs through DNA-methylation analysis. We also identified genetic driver mutations specific to different cells of origin which are related to clinical outcome. Based on our findings we propose a new model of PanNET origin and progression.

## Results

### DNA-methylation signature of PanNETs showed similarities to α- and β-cells.

We analysed the methylomes of 125 PanNETs and of isolated normal α- and β-cells[20] to determine which of these is the cell of origin (cohort details are summarised in Table 1 and reported in Supplementary Data 1). All samples were processed on the HumMeth450 BeadChip platform (Illumina HM450). We carried out the assays for 45 primary PanNETs (UCL and UB cohorts), while DNA-methylation data for the additional 80 samples (ICGC cohort) were produced by the International Cancer Genome Consortium (ICGC). A flow chart of the analysis performed is provided in Supplementary Fig. 1. The comparison between sorted normal α- ($n = 2$) and β-cells ($n = 2$) resulted in 2703 differentially methylated CpG sites (adj. $p$ value < 0.001 and more than 20% difference between the mean β-values of each group—$|\Delta\beta| > 0.2$, Supplementary Data 2), 2131 of them were retained, after filtering and normalization, in the tumour

### Table 1 Patient cohort used for methylome analysis (cohort 1).

| | UniBern cohort | UCL cohort | ICGC cohort | Total |
|---|---|---|---|---|
| **Total number of patients** | 26 | 19 | 80 | 125 |
| **Sex** | | | | |
| Females | 10 | 12 | 31 | 53 |
| Males | 16 | 7 | 49 | 72 |
| **Grade (WHO 2010)** | | | | |
| G1 | 6 | 8 | 32 | 46 |
| G2 | 20 | 11 | 48 | 79 |
| **Tumour Stage (AJCC 8th ed.)** | | | | |
| T1 | 3 | 3 | 10 | 16 |
| T2 | 8 | 10 | 18 | 36 |
| T3 | 15 | 4 | 26 | 45 |
| T4 | 0 | 0 | 26 | 26 |
| No data | 0 | 2 | 0 | 2 |
| **N stage** | | | | |
| N0 | 10 | 7 | 42 | 57 |
| N1 | 13 | 4 | 36 | 52 |
| No data | 3 | 8 | 2 | 13 |
| **M stage** | | | | |
| M0 | 12 | 2 | 67 | 80 |
| M1 | 10 | 1 | 13 | 24 |
| No data | 4 | 16 | 0 | 21 |
| **DAXX/ATRX** | | | | |
| Wild-type | 8 | 11 | 53 | 72 |
| Mutated | 18 | 7 | 27 | 52 |
| Not applicable | 0 | 1 | 0 | 1 |
| **ALT** | | | | |
| Negative | 8 | 11 | 42 | 61 |
| Positive | 16 | 3 | 29 | 48 |
| Not Applicable | 2 | 5 | 9 | 16 |
| **MEN1 (somatic mutations)** | | | | |
| Wild-type | 12 | 13 | 49 | 74 |
| Mutated | 14 | 6 | 31 | 51 |
| **Hormone functionality** | | | | |
| F-PanNETs | 8 | 2 | 12 | 22 |
| NF-PanNETs | 18 | 0 | 68 | 86 |
| Not Applicable | 0 | 17 | 0 | 17 |
| **Syndromic—MEN1** | 0 | 2 | 3 | 5 |
| **Median follow-up time (mths)** | 83 | 40 | 37.5 | 37.9 |

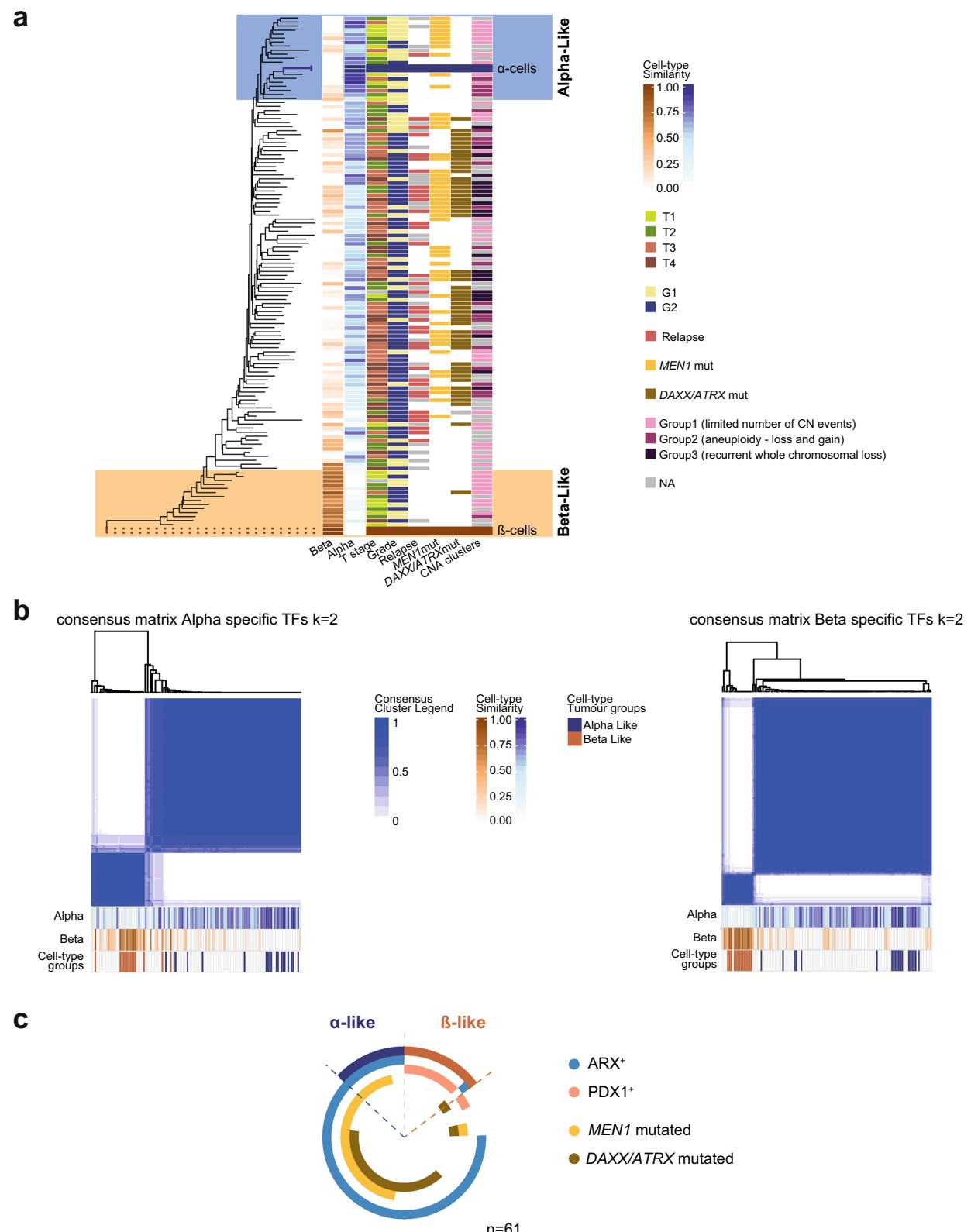

DNA-methylation table (including the values of methylation for 363,665 CpG sites and 125 samples). Phyloepigenetic analysis of the 2131 identified CpG sites using either normal β- or normal α-cells as roots, indicated clear hierarchical relationships between normal cells and tumour specimens (Fig. 1a, Supplementary Fig. 2a). We defined PanNETs that grouped together with normal α- or β-cells as α-like or β-like PanNETs, respectively (Fig. 1a and

Supplementary Data 1). We compared these phyloepigenetic groups to cell type similarity obtained from cell-type deconvolution analysis: using the algorithm of Houseman et al.[21] (a method similar to regression calibration), we inferred the contribution of DNA-methylation profiles of sorted normal pancreatic cell types (α, β, acinar, ductal[20], fibroblastic pancreatic cells[22]) and hematopoietic cells[23] to the methylation profiles of

**Fig. 1 PanNET methylomes resemble distinct endocrine cell lineages. a** Phyloepigenetic analysis of PanNET ($n = 125$) and normal α- and β-cell samples ($n = 2$ for each cell type). A rooted tree was created with an arbitrary chosen β-sample as the root and selecting the differentially methylated CpGs between sorted normal α- and β-cell samples ($n = 2131$, adj. $p$ value < 0.001 and $|\Delta \beta| > 0.2$). Blue and orange scales show the degree of similarity of each PanNET sample to α- and β-cells, respectively. The degree of similarity was calculated via DNA-methylation cell-type deconvolution comparing tumour methylomes to sorted normal α- and β-cell methylomes. Each line represents a patient. Clinical and molecular features for each sample are indicated. **b** Consensus clustering of the 125 PanNETs according to the α- and β-cell-specific TF-checkpoint methylation sites (49 CpGs for β-cells and 51 CpGs for α-cells). The k mean value was set to 2. Consensus cluster correlation is indicated according to the blue scale as depicted. Fractions of similarity to α- and β-cells for each sample are reported at the bottom of each matrix according to the blue and orange scales. Similarly, the cell-type-specific tumour groups are reported for each sample in blue (α-like), orange (β-like), and white (intermediate). **c** Schematic overview of the PanNET subtypes. Doughnut chart for all samples showing cell-specific PanNET subtypes, ARX and PDX1 expression, *MEN1* and *DAXX/ATRX* mutations.

---

| Table 2 Specific cell-type TF checkpoints. | |
|---|---|
| **Alpha** | **Beta** |
| | *MAFA* |
| *FEV* | *PDX1* |
| *IRX2* | *SMAD9* |
| *LDB2* | *CDKN1C* |
| *MAFB* | *TFCP2L1* |
| *PGR* | *SIX3* |
| *PTGER3* | *SIX2* |
| *RFX6* | *MNX1* |
| | *BMP5* |

PanNETs (Supplementary Fig. 2b). Notably, phyloepigenies of tumours that were close to normal α/β-cells showed similarity >65% and an average of 72 and 73% of similarity to α and β-cells, respectively (SD for α-like ±15% and for β-like ±9%, Supplementary Data 1). T-distributed stochastic neighbour embedding (*t*-SNE) analysis, using the identified 2131 CpGs and including normal cells and PanNET samples, showed a consistent segregation, according to the groups defined via similarity scores and phyloepigeny analysis (Supplementary Fig. 2c). These results showed at least two groups of tumours with clear and high similarity to either α- or β-cells (Fig. 1a, Supplementary Fig. 2a and Supplementary Fig. 2c). The majority of tumours, however, revealed an intermediate differentiation profile. Intermediate tumours clustered between α- and β-cells and showed weak similarities to α- (as average $51 \pm 20\%$ of similarity) and very weak similarity to β-cells (as average $14 \pm 16\%$ of similarity), respectively.

**Regulatory sites of α- and β-cell-specific transcription factor checkpoints were differentially methylated**. We used previously published data to select 10 α- and 10 β-cell-type associated[24] TF checkpoints (genes reported to function as RNA polymerase II—regulating DNA-binding transcription factors and included in the TFcheckpoint database[25]). As methylation of enhancers and regulatory chromatin states are important elements of epigenetic guidance of cell states, these regions were selected based on integrated ChIP-seq, DNA-methylation and ATAC-seq experiments of normal islets[26]. After selection of regulatory regions on autosomal chromosomes, probes retained for the analysis were related to 7 α-specific and 9 β-specific TF checkpoints (listed in Table 2). Specifically, the selection resulted in 49 CpGs for the α-cell TF checkpoints and 51 CpGs for the β-cell TF checkpoints (list and relative features of the selected CpG sites are reported in Supplementary Data 3 and 4). Consensus clustering of the 125 PanNETs according to each cell-specific TF-checkpoint signature (k-means = 2, reflecting one cell-type enriched cluster and one nonspecific cluster) showed clear and consistent cluster formation. The β-specific cluster separated very clearly and included all the β-like tumours. Interestingly, the α-specific cluster showed a

broader inclusion of tumours. While the α-like (phyloepigenies with >65% of similarities to alpha cells) tumours clustered mainly close to each other on one extreme, intermediate PanNETs were included to this group as well but progressively distant from the α-like tumours (Fig. 1b).

**Alpha-, intermediate and β-like PanNETs have distinct genetic aberrations**. We screened *MEN1*, *DAXX* and *ATRX* genes for mutations and we classified *DAXX/ATRX* mutated samples, those tumours which are mutated in either or both the genes. Within the β-like PanNETs 13 of 14 were *MEN1/DAXX/ATRX* wild-type. Alpha-like PanNETs were enriched for tumours with only *MEN1* mutations (11/19), 8/19 α-like PanNETs were *MEN1/DAXX/ATRX* wild-type and none harboured *MEN1* and *DAXX/ATRX* mutations (Fig. 1a and Supplementary Data 1). Sixty-seven percent (62/92) of intermediate PanNETs harboured mutations in *MEN1* and/or *DAXX/ATRX* genes (intermediate-ADM) while the rest were wild-type (intermediate-WT) (Fig. 1a and Supplementary Data 1).

Inferred copy-number aberrations (CNAs) from HM450 signal intensities stratified the tumours into three subtypes (Supplementary Fig. 3e). Copy-number aberration (CNA) group 1 had few copy-number events. Group 2 included samples with gains and losses. CNA group 3 showed the largest number of CNAs, predominantly with loss of multiple chromosomes (Supplementary Fig. 3e). Notably, α- and β-like tumours presented few copy-number events while the intermediate PanNETs showed increasingly copy-number events (Fig. 1a).

**DNA-methylation signatures associated with cell of origin correlated with transcription factor expression**. To confirm the methylation-based cell lineage similarity, we performed immunostaining for ARX and PDX1 on a subset of samples for which formalin-fixed paraffin-embedded tissue was available ($n = 61$, Supplementary Data 1; for $n = 39$ cases also glucagon and insulin expression was assessed; results are reported in Supplementary Data 1). Nuclear positivity for ARX and PDX1 was present in the expected distribution in normal islets (Supplementary Fig. 4). We identified ARX positivity in 47/61 PanNETs (Supplementary Data 1 and Fig. 1c). As expected, the majority of insulinomas expressed PDX1, except one malignant and one *DAXX/ATRX* mutated insulinomas, which showed nuclear positivity for ARX but not for PDX1 (Supplementary Data 1 and Fig. 1c). None of the 61 tumours were double positive and only four PanNETs were double negative (Supplementary Data 1 and Fig. 1c).

Immunohistochemistry confirmed the same group segregation as was determined by the methylation analysis: (1) α-like tumours expressed ARX (Fig. 1c, Supplementary Data 1), (2) β-like tumours expressed PDX1, with only one sample positive for ARX and PDX1 negative (Fig. 1c, Supplementary Data 1), (3) 38/44 (86%) intermediate PanNETs were positive for ARX and only two cases were PDX1 positive and ARX negative. A minority of

## Table 3 Validation cohort (cohort 2).

|  | Patient cohort |
|---|---|
| **Total number of patients** | 65 |
| **Sex** | |
| Females | 28 |
| Males | 31 |
| No data | 6 |
| **Grade (WHO 2017)** | |
| G1 | 34 |
| G2 | 25 |
| No data | 6 |
| **Tumour stage (AJCC 8th ed.)** | |
| T1 | 29 |
| T2 | 11 |
| T3 | 15 |
| T4 | 1 |
| No data | 9 |
| **N stage** | |
| N0 | 30 |
| N1 | 15 |
| No data | 20 |
| **M stage** | |
| M0 | 40 |
| M1 | 11 |
| No data | 14 |
| **DAXX/ATRX** | |
| Positive | 51 |
| Negative | 12 |
| Not applicable | 2 |
| **Hormone functionality** | |
| F-PanNETs | 20 |
| NF-PanNETs | 44 |
| Syndromic—MEN1[b] | 1 |
| **ARX** | |
| Positive | 34 |
| Negative | 31 |
| **PDX1** | |
| Positive | 21 |
| Negative | 44 |
| **Median follow-up time (mths)** | 55 |

intermediate tumours (4/44, 9%) were negative for both PDX1 and ARX (Fig. 1c, Supplementary Data 1).

These results indicate that cell type assigned by DNA methylation, corresponded to the cell type assigned by TF expression.

**PDX1 and ARX immunopositivity correlates with mutation spectrum of PanNETs in an independent cohort**. To validate the correlation of putative cell of origin to mutational status, we performed immunostaining for ARX, PDX1, insulin, glucagon and DAXX/ATRX on an independent cohort of 65 G1/G2 PanNETs (cohort details are summarised in Table 3 and reported in Supplementary Data 5). We scored nuclear positivity for ARX in 34 samples (52%), for PDX1 in 21 samples (32%), 3 tumours (5%) showed strong double positivity and 13 (20%) were negative for both TFs (Figs. 2a, b). Notably, all the DAXX/ATRX-negative samples expressed ARX, confirming the α-like tumour susceptibility for these mutations (Figs. 1a, 2c). While only a subset of ARX+ tumours secreted glucagon (n = 7, 20% of all the ARX+ tumours, Fig. 2c), almost all the PDX1+ tumours produced insulin (n = 18, 95% of all PDX1+ tumours, Fig. 2c).

We observed that advanced stage PanNETs either expressed ARX or none of the TFs and often showed DAXX/ATRX loss (Fig. 2c, Supplementary Fig. 5a and Supplementary Data 5). Tumours positive for PDX1 had the lowest risk of relapse

(Supplementary Fig. 5b and c). However, DAXX/ATRX status did not improve stratification of ARX+ samples for risk of relapse (Supplementary Fig. 5b).

**Epigenetic differentiation status determines clinical outcome of PanNETs**. In order to determine clinical utility, we reviewed clinical outcome and prognosis for α-like, β-like and intermediate tumours (from cohort 1). Fourteen of 19 α-like PanNETs were G1 and 15/19 were of low stage (T1 or T2, Fig. 1a and Supplementary Data 1). Similarly, 7/14 β-like PanNETs were G1 and 11/13 (1 without data) were T1 or T2 (Fig. 1a and Supplementary Data 1). Only one patient with an α-like tumour and none of the patients with a β-like tumour had relapse (Figs. 1a, 3a and Supplementary Data 1). Tumours that belonged to the intermediate phyloepigenies, were instead enriched for high stage (65/91 were T3 or T4, 1 without data) and higher grade (67/92 are G2) PanNETs and had increased relapse risk (35/67, 25 had not available data—Figs. 1a, 3a and Supplementary Data 1). Intermediate tumours showed generally higher Ki-67 positivity (Supplementary Fig. 6). Disease free survival in intermediate tumours was significantly shorter than in α-like and β-like tumours (p < 0.001, Fig. 3a). To assess if epigenetic profile can stratify patients better than TF and DAXX/ATRX expression we performed disease free survival analysis on all samples where DNA-methylation and TF expression data were available. We determined time to relapse, using either expression of ARX, PDX1 and DAXX/ATRX or the epigenetic status as discriminant (Fig. 3b, c). Epigenetic profiles could predict more accurately the risk of relapse as shown in Fig. 3b, c and Supplementary Fig. 5c.

Further, to identify a tumour dependent signature, suitable for stratification in clinical practice, we have compared the three cell-type specific clusters, α-like, β-like and intermediate. The comparison revealed 6364 unique differentially methylated sites (Supplementary Data 6–8; adj. p value < 0.001 and |Δβ| > 0.2). Consensus clustering of the 6364 CpG sites showed the maximal stability for k = 4 (Supplementary Fig. 7a). The existence of α-like and β-like clusters was confirmed (Fig. 3d and Supplementary Data 1). The intermediate subgroup was divided into two further groups, one enriched for DAXX/ATRX/MEN1 mutations, intermediate-ADM (61/76 had mutations in at least one of the three genes; Fig. 3d and Supplementary Data 1) and the other enriched for DAXX/ATRX/MEN1 wild-type samples, intermediate-WT (15/16, only 1 DAXX/ATRX mutated; Fig. 3d and Supplementary Data 1). PanNETs included in the intermediate-ADM and intermediate-WT showed comparable risk to relapse (Supplementary Fig. 7b). To prove the stability of the signature, we repeated the analysis including 32 new G1/G2 PanNETs, for which DNA-methylation data were public available[7] (Supplementary Data 9). After filtering and normalization processes, 6359/6364 of the previously identified differentially methylated sites were used in the consensus clustering algorithm. The results confirmed the 4 PanNET subgroups with similar relapse risks to the previous analysis (Supplementary Fig. 7c–e).

## Discussion

This study identified at least two cells of origin for PanNETs (α-like/β-like) and demonstrated that DNA-methylation analysis can discriminate α-like, β-like and intermediate (-ADM and -WT) PanNETs. Additionally, this study identified that relapses and metastases occurred most commonly in the intermediate (-ADM and -WT) PanNET groups.

Phyloepigenetic analysis of PanNETs, according to differentially methylated CpG sites between normal α- and β- cells, showed two clusters around normal α- and β- cells, which we named α-like and β-like. PanNET sample segregation was

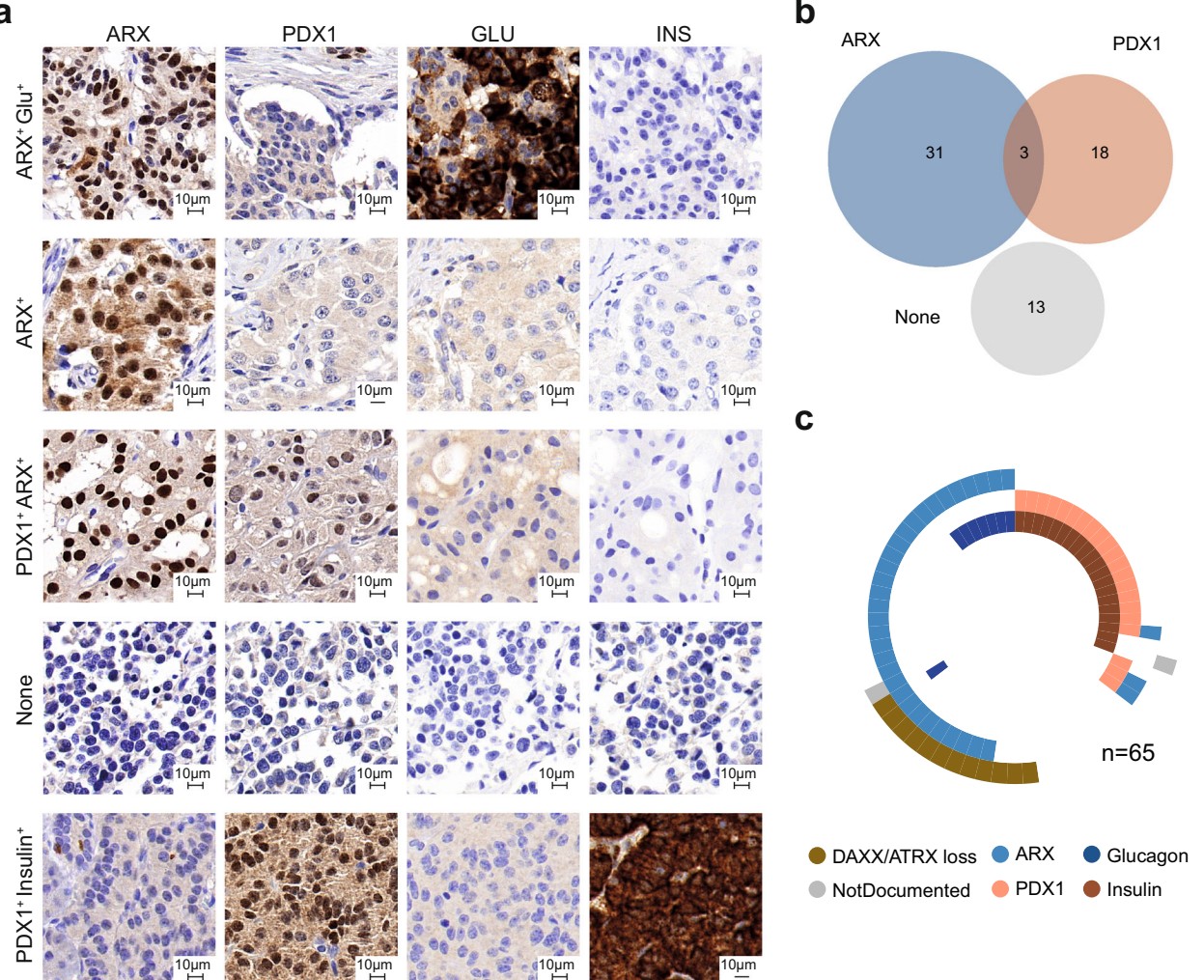

**Fig. 2 Expression of ARX, PDX1, insulin, and glucagon can be used as surrogate for defining tumour cell-type specificity. a** Immunostaining for the different subtypes of PanNETs. From the top to the bottom: tumour positive for ARX and glucagon; tumour positive for ARX; double positive tumour for PDX1 and ARX; double negative tumour for ARX and PDX1; tumour positive for PDX1 and insulin. For the TFs only nuclear staining was considered for scoring. **b** Venn diagram displaying the total number of samples positive for ARX and/or PDX1or negative for both the transcription factors. **c** Schematic overview of the PanNET subtypes in cohort 2. Doughnut chart for all samples showing glucagon, insulin, PDX1 and ARX positivity as well as loss of DAXX/ATRX expression.

consistent in the *t*-SNE analysis. Using a similar approach, in a recent study, methylation-based subclasses of colorectal cancer were explained as clonal amplifications of one specific epigenotype, confirmed by the enrichment of different mutations in the particular subtypes and by the investigation of methylation signatures in serial tumour xenografts and derived spheroids[18]. Similarly, in medulloblastoma subtypes with distinct developmental origin, DNA-methylation signatures were able to stratify tumours according to the specific subtype, to provide information about the cell of origin and also to demonstrate acquired epigenetic changes during tumour progression[14,27,28]. Even if in the aforementioned studies tumours were not directly compared to the putative normal cells of origin, the epigenetic and mutational landscape of the malignancies reflected the tumour classes and cells of origin defined via in vivo studies. Altogether the data confirm the valuable use of DNA-methylation profiles for the identification of tumour cells of origin.

Cell-type deconvolution analysis of methylome data from ductal, acinar, inflammatory and pancreatic stromal cells together with α- and β- cells found that α-like and β-like PanNETs are largely similar (at ~73%) to α- and β- cells, respectively. This

figure is comparable to other studies: Houseman and Ince[29] demonstrated the application of their algorithm to estimate normal cell proportions in breast cancer heterogeneous tissues. Similarly, cell-type deconvolution analysis based on DNA methylation for mantel cell lymphoma (MCL), showed a minimum of similarity between MCL and B cell samples of 40%[30]. When a similar algorithm (L1-regularized logistic regression) is used to classify cancers of unknown origin (CUPs) based on DNA methylation, a probability >30% was used to ascribe CUPs to a specific tumour type[31]. The deconvolution of the methylation estimates for PanNETs identified two important aspects of the tumour methylomes: that the results were not influenced by the composition of non-tumoral cells within the PanNET samples (Supplementary Fig. 2b) and that the composition of early stage α-like and β-like PanNETs is abundant for either α- or β-cells (Fig. 1a).

Analysis of methylation status of α- and β-cell-specific TF checkpoints regulating differentiation might help to better identify the intermediate groups of PanNET. While β-like samples separated very clearly according to β-cell TF-checkpoint methylation sites, α-cell similarity decreased gradually among the

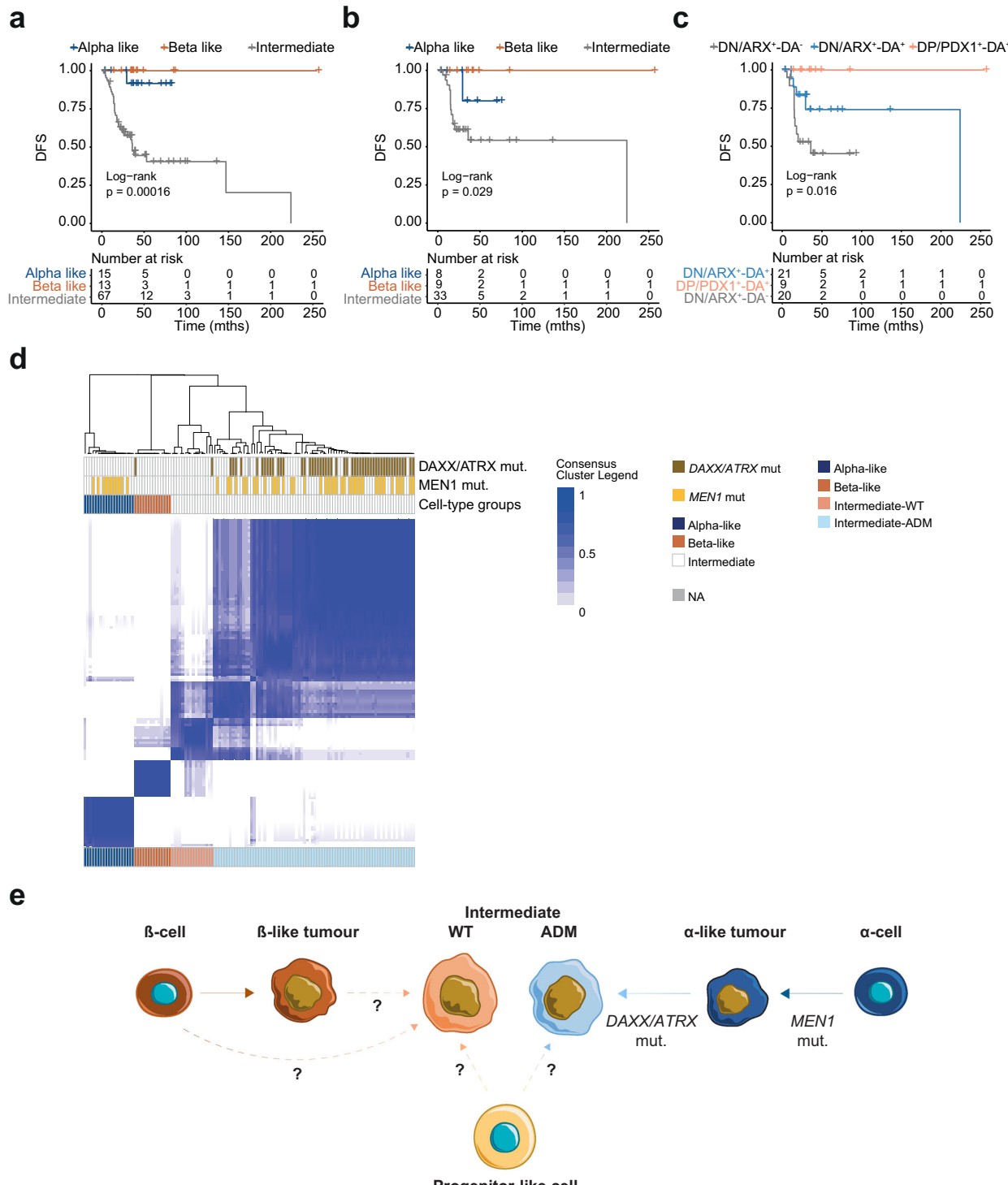

remaining samples. Seventy of seventy-six intermediate-ADM PanNETs clustered together with the α-like tumours (Supplementary Data 1). Twelve of sixteen intermediate-WT PanNETs did cluster neither with α-like nor with β-like tumours (Supplementary Data 1). Additionally, intermediate-ADM PanNETs were on average more similar to α- (53 ± 17%) than β-cells (12 ± 13%). Ninety-five percent of the intermediate-ADM tumours were positive for ARX (36/38) and none for PDX1. This might indicate that intermediate-ADM PanNETs are more related to α-cells rather than β-cells. The PanNET methylation subgroups that we have identified reflect the groups A (tumours that most likely originate form α-cells), B (tumours that most likely originate form β-cells) and C (tumours with intermediate phenotype) recently described by Cejas et al., via the analysis of super-enhancer signatures[8]. Comparable to our study, the analysis of TF expression via IHC performed by Cejas et al. revealed mutually exclusive expression of either ARX or PDX1 in the majority of the cases. Additionally, while PDX1 expression appears to be specific for benign and low stage tumours, ARX expression is retained at early and at advanced stages and only DAXX/ATRX status provides more information about stage and risk of disease progression[8].

**Fig. 3 Epigenetic differentiation status defines clinically different PanNETs and draws two possible ways of PanNET evolution. a** Kaplan–Meier disease free survival of 95 patients (cohort 1) stratified according to cell-type specific methylation groups (α-like in blue, β-like in orange and intermediate in grey). Intermediate tumours have higher risk of relapse compared to α- and β-like tumours (p value = 0.00016). In **b** and **c** disease free survival of a subset of patients of cohort 1 (n = 50) with available data for both DNA-methylation profile and PDX1, ARX, and DAXX/ATRX IHC. In **b** patient stratification according to cell-type-specific methylation groups and in **c** according to PDX1, ARX, and DAXX/ATRX IHC (in grey patients double negative for PDX1 and ARX or ARX positive and negative for DAXX/ATRX, in light-blue patients double negative for PDX1 and ARX or ARX positive and DAXX/ATRX positive, in light-orange patients double positive for PDX1 and ARX or PDX1 positive and DAXX/ATRX positive). **d** Consensus clustering of the 125 PanNETs according to the 6364 differentially methylated sites between α-like, β-like and intermediate tumours (adj. p value < 0.001 and |Δβ| > 0.2). Cluster stability was reached for k = 4 (see Supplementary Fig. 7a). Consensus cluster correlation is indicated according to the blue scale as depicted. Each column represents one PanNET sample. Tumour mutations and cell-type subgroups are indicated according to the reported colours. **e** Progression model hypothesis based on epigenetic and genetic evolution: α-like tumours originate from α-cells upon *MEN1* inactivation, the progression to intermediate-ADM tumours is enhanced by secondary events, including loss of DAXX/ATRX and chromosomal instability with recurrent LOH and activation of ALT. These events are associated to a gradual loss of differentiation. Beta-like tumours originate from β-cells upon different genetic event and they are mainly insulinoma, usually indolent. However, based on our data it is not possible to exclude that intermediate-WT PanNETs originate directly from β-cells or from a progenitor cell. Similarly we cannot exclude that intermediate-ADM may originate from a precursor cell as well. Progression supported by genetic and epigenetic data is depicted by full lines, while hypothetical progression is depicted by dotted lines.

We identified a clear correlation between driver mutation status and epigenetic profiles across all PanNETs. Alpha-like PanNETs and intermediate-ADM PanNETs harboured *MEN1* mutations, this is supported by the results of Chan et al. and Cejas et al.[7,8]. Fifty-eight percent of the clinically indolent α-like PanNETs were characterized by *MEN1* mutations only. *MEN1* inactivation, is an early event in PanNET progression[32,33], which enhances endocrine cell proliferation[34], hence it might be a tumour initiating event for the α-like PanNETs and for the intermediate-ADM. In turn Intermediate-ADM PanNETs might progress upon acquisition of DAXX/ATRX inactivation. In addition, CNA increased from α-like and β-like PanNETs to intermediate-ADM tumours. All these data together suggest a potential progression from α-like to intermediate-ADM PanNET. We cannot exclude, however, that the two intermediate PanNET clusters might originate from putative endocrine precursor cells.

During islet cell development, ARX is already expressed in endocrine precursors[11], furthermore different endocrine cell types, as α- and γ-cells share ARX expression[24,35]. Conversely PDX1 expression is restricted to differentiated β-cells, within the endocrine lineages[12]. Of the intermediate-WT tumours for which IHC data were available (n = 6), two were positive for PDX1, two for ARX and two were negative for both TFs. Additionally, similarly to the β-like tumours, intermediate-WT PanNETs showed only few copy-number events (group 1, Supplementary Data 1). Currently our data only weakly support progression from β-like to intermediate-WT PanNETs, nevertheless this possibility cannot be excluded.

While the vast majority of β-like PanNETs expressed PDX1 and insulin, the two malignant (N1 and/or M1) insulinomas of our cohort, showed one intermediate-ADM (*DAXX/ATRX* mutated) and one α-like (*DAXX/ATRX* wild-type) methylation signatures (Supplementary Data 1). Only for the intermediate-ADM insulinoma, data on PDX1 and ARX were available; it resulted positive for ARX, and negative for PDX1 expression. Accordingly, the only malignant insulinoma of the Chan et al. cohort was mutated for *ATRX* and clustered with the intermediate-ADM tumours. In line with this observation, a recent study including 37 sporadic insulinomas (35 primary and 2 liver metastases) showed that all the five insulinomas which metastasized were ARX positive and 4/5 had ALT activation (3 primary and 2 liver metastases)[36]. These data suggest that malignant insulinomas may arise from α-cells or stem-cells rather than β-cells. Under certain conditions and stimuli α-cells are able to trans-differentiate into β-cells[37–41]. Additionally, α-cell-specific *Men1* knockout in mice leads to the development of glucagonomas, which evolve into mixed glucagonoma/insulinoma to ultimately become insulinomas, possibly via trans-differentiation of

the *Men1*-deleted α-cells[42,43]. Beta-like and/or PDX1 positive tumours are strongly enriched for benign insulinomas in both the first and second cohort. They generally show no mutations in any of the most commonly mutated genes for PanNETs (*DAXX, ATRX* and *MEN1*). These data confirmed the genetic difference between non-functioning tumours and insulinomas[5,44–46]. Of the thirteen wild-type PanNETs obtained from Chan et al.[7], five were included in the β-like cluster (two insulinomas and three NF-PanNETs), six in the intermediate-WT cluster and only two in the intermediate-ADM cluster (Supplementary Data 9). All the ADM-mutant PanNETs were included in the intermediate-ADM cluster (Supplementary Data 9). These data and the few β-like NF-PanNETs included in the cohort 1, demonstrate that occasionally also NF-PanNETs might originate from β-cells.

Our methylation data support the possibility of two evolutionary pathways for PanNET development, originating from α- and β-cells (Fig. 3e). Beta-like PanNETs usually manifest as insulinomas. The α-like PanNETs are susceptible to *MEN1* mutations in early tumorigenesis. Tumour progression occurs upon *DAXX/ATRX* mutations, coupled with ALT activation and a characteristic CNA profile[4,47–49]. Progressive loss of differentiation might further predispose the tumours to enhanced proliferation and higher cell plasticity (Fig. 3e). Alternatively or additionally to this dedifferentiation model, potential endocrine progenitor cells might be the cell of origin of intermediate tumours (Fig. 3e). While CNA, genetic background and epigenetic suggest a progression from α-cells to α-like tumours first and then to intermediate-ADM, we do not have the same evidence supporting an evolution from β-like tumours to intermediate-WT, even if we cannot completely exclude it. In fact, while a progression from β-like benign insulinoma to aggressive intermediate-WT tumours is, based on clinical evidences, very unlikely; a direct origin of these tumours from normal adult β-cells or from a common progenitor is more plausible. Expression of cell type TFs such PDX1 and ARX is equally distributed in intermediate-WT however DNA-methylation profile indicates similarity with β-cells (Fig. 3d and Supplementary Fig. 7d).

We acknowledge that our study is based on static observations taken at one timepoint in each tumour's development. Sequential sampling or in vivo experiments would be able to determine real time tumour evolution and would be able to address the unanswered questions we propose here.

Clinically α-like, β-like and intermediate tumours have different outcomes. While α-like and β-like tumours are indolent, intermediate tumours (-ADM and -WT) have high risk of relapse. Interestingly, DAXX/ATRX status alone is not sufficient to discriminate between ARX+ with high and low risk of relapse. Indeed, in the intermediate-WT group we observed the presence of ARX+

tumours, with high risk of relapse. Only the DNA-methylation profile is able to separate ARX[+] PanNET with low risk from high risk of relapse. The intermediate groups of PanNET are characterized by high risk of relapse but are molecularly different (ADM and WT) (Fig. 3d and Supplementary Fig. 7d). Further work is needed to assess whether intermediate-ADM and intermediate-WT aggressive PanNETs have different origins or respond differently to therapies.

DNA-methylation analysis clearly bears advantages over ChIP-seq assays, it is easily performed on diagnostic routine FFPE specimens. We foresee a potential clinical use of epigenetic profiling for PanNETs similar to CNS tumor classification, able to define clinically relevant groups[28,50,51]. In addition, DNA methylation remains stable in circulating tumor cells (CTCs) and cell-free DNA (cfDNA) (reviewed in ref. [52]), hence might be applied to monitor progression in individual patients in a non-invasive liquid biopsy. DNA-methylation profiles can help in identifying patients with risk of relapse and we envision that in the future can help in predicting therapy response.

In conclusion DNA-methylation analysis could be easily implemented in clinical practice to identify patients with high relapse risk and those which might benefit from adjuvant therapy.

## Methods

**Patient cohorts.** A cohort of primary PanNETs was assembled from two international centres; 19 samples previously analyzed[53] from UCL Cancer Institute (London, UK) and 26 from Institute of Pathology (Bern, Switzerland). All cases were classified according to WHO 2017 criteria[54]. TNM staging is based on the 8th edition UICC/AJCC. Inclusion criteria were histopathologic diagnosis of well-differentiated G1/G2 PanNETs, availability of tissue material and sufficient tumour purity (>70%). Fourteen of 45 tumour samples (31%) were classified as G1, 31/45 (69%) as G2 (Table 1). Two samples derived from MEN1-patients. All analyses were performed on Formalin-Fixed Paraffin-Embedded (FFPE) tissue specimens obtained from routine pathological work-up. Additional clinico-pathological characteristics are reported in Supplementary Data 1. The study on this cohort was approved by the local Research Ethics Committees (Bern: number 105/2015; London: number 09/H0722/27). Patients recruited through the Inselspital with operation data before 2015 did not object to use of data and biological material for research, all patients included after 2015 did sign a written broad institutional consent. Informed written consent for patients recruited through the Royal Free Hospital Neuroendocrine Unit was obtained before entering the study. All samples were fully anonymised.

Clinical and molecular data of 80 PanNETs were provided by the International Cancer Genome Consortium (ICGC, https://icgc.org/, projects: PAEN-AU and PAEN-IT). Cohort features are summarised in Table 1 and Supplementary Data 1. Genomic and molecular analysis of the samples were performed within Scarpa et al. study[47].

As immunohistochemical validation cohort, 65 PanNET samples were assembled at the Institute of Pathology (Bern, Switzerland). All cases were reclassified according to WHO 2017 criteria[54]. TNM staging is based on the 8th edition UICC/AJCC[55]. As for the previous cohorts, only well-differentiated G1/G2 PanNETs were included. Cohort features are summarised in Table 3 and extensively reported in Supplementary Data 5. The study on this cohort was performed according to the protocol approved by the ethical committee (number 105/2015) according to Swiss Human research Act. Patients with operation data before 2015 did not object to use of data and biological material for research, all patients included after 2015 did sign a written broad institutional consent.

A flow chart of the analysis performed on the 2 cohorts is provided in Supplementary Fig. 1.

**Immunohistochemistry (IHC) and telomeric-fluorescence in situ hybridization (Telo-FISH).** All samples were assessed for DAXX (1:40, anti-DAXX, polyclonal rabbit; Sigma–Aldrich, St. Gallen, Switzerland) and ATRX (1:400, anti-ATRX, polyclonal rabbit; Sigma–Aldrich) expression via IHC and ALT activation via Telo-FISH on 2.5 µm sections prepared from a tissue microarray (TMA - Bern cohort and validation cohort) or whole block sections (London cohort). For FISH, sections were deparaffinised and rehydrated, slides were then boiled for 30 min in citrate buffer, pH 7.2, and incubated for 30 min in 2 × standard saline citrate and 0.05% Tween 20. A peptide nucleic acid (PNA) probe (telC-Alexa488; Panagene, Daejeon, Korea) was diluted (1:10) in 70% formamide, 10 nmol/L Tris, pH 7.5. One drop of PNA solution was spotted on hydrophobic gel bond film and mounted on a glass slide. Samples were denatured at 85° for 4 min and incubated for 2 h at room temperature in the dark. Following slides were washed in 60% formamide 10 nmol/L Tris, pH 7.5, for 5 min and 2 × standard saline citrate-Tween 20. Anti-promyelocytic leukemia (PML) (antibody PG-M3; Santa Cruz, Heidelberg,

Germany) 1:100 was incubated for 1 h at room temperature and the secondary antibody (goat-anti-mouse Alexa568; Cell Signaling, Danvers, MA) was diluted 1:500 and incubated for 1 h at room temperature in a dark chamber. One percent 4′,6-diamidino-2-phenylindole solution was incubated for 3 min at room temperature. FISH was evaluated using an Olympus VS 110 Fluorescent Scanner (Olympus, Volkestwil, Switzerland).

For the immunostainings, antigen retrieval for DAXX was performed by heating citrate buffer at 100° for 30 min and for ATRX in 95° Tris buffer for 40 min. Primary antibody was incubated for 30 min at the specified dilutions. Visualization was performed using the avidin-biotin complex method. For both DAXX and ATRX scoring, only nuclear protein staining was considered positive.

Similarly, 2.5 µm sections from TMAs or whole blocks were used for ARX (1:1500, R&D Systems, sheep, AF7068), PDX1 (1:100, R&D Systems, mouse, MAB2419), insulin (1:12000, Sigma–Aldrich, mouse, I-2018) and glucagon (1:20000, Sigma–Aldrich, mouse, G-2654) immunostainings. Antigen retrieval was performed by heating Tris30 buffer at 95 °C for 30 min. The primary antibody was incubated for 30 min at the specified dilutions. Visualization was performed using Bond Polymer Refine Detection kit, using DAB as chromogen (3,3′-Diaminobenzidine).

Samples showing single cell positivity of any of the TFs were finally scored as negative, most of the cases were indeed strong positive for one of the two TFs with only few exceptions (single cell positivity is reported for each sample in Supplementary Data 1 and 5). Only strong positivity for insulin and glucagon was considered for classification of the tumours as hormone producing PanNETs. Nuclear positivity for ARX and PDX1 was first assessed on normal islets, proving the selectivity for endocrine α- and β-cells (Supplementary Fig. 4).

The immunostaining for all antigens was performed on an automated staining system (Leica Bond RX; Leica Biosystems, Nunningen, Switzerland).

**DNA-methylation analysis.** We extracted DNA from FFPE tissues according to manufacturer recommendations (QIAamp DNA minikit, Qiagen). Serial sections were cut and macrodissected using a razor blade upon histological evaluation (5 × 6 µm), to make sure to achieve >70% tumour purity. We assessed DNA quality using the Illumina FFPE QC Kit. Ligation of FFPE DNA and bisulphite conversion were performed as described[56]. Efficiency of bisulphite conversion was confirmed by quantitative PCR as previously shown[56]. Converted DNA was processed on the HumMeth450 BeadChip (Illumina HM450). We analysed all the DNA-methylation data included in this study using the ChAMP pipeline (v2.12.4, minfi method was used for raw data loading)[57–59]. Filtering was performed as implemented in the ChAMP pipeline[60]. Only probes located on autosomal chromosomes were retained. Type II probe bias was corrected using the BMIQ method as part of the ChAMP pipeline[61]. Batch correction was performed using the ComBat algorithm as part of the ChAMP pipeline[62,63].

We identified differentially methylated sites between normal α− and β−cells, as well as between α−like β−like and intermediate PanNETs, according to the ChAMP pipeline[64,65]. To build phyloepigenetic trees, distances between samples were calculated according to pearson correlation and the neighbour-joining tree estimation was used within the ape (v5.3) R package[66]. The tSNE approach was performed as implemented in the R package tsne using the following parameters: perplexity = 50, max_iter = 5000[67].

We used the RnBeads pipeline[68] for calculating cell-type contribution according to the Houseman et al. method[21]. Sorted normal hematopoietic cell-type data (granulocytes, CD4 + and CD8 + T cells, CD14 + monocytes, CD19 + B cells, CD56 + natural killer cells, neutrophils and eosinophils) were downloaded from Reinius et al.[23] (Gene Expression Omnibus, http://www.ncbi.nlm.nih.gov/geo/, accession number: GSE35069). Sorted acinar, duct, alpha and beta pancreatic cell DNA-methylation profiles were downloaded from Jäkel et al.[20] (European Genome-Phenome Archive, https://ega-archive.org/, accession number: EGAS00001002533). DNA-methylation data for sorted normal pancreatic fibroblast cells were downloaded from Xiao et al.[22] (Gene Expression Omnibus, http://www.ncbi.nlm.nih.gov/geo/, accession number: GSE80369). Marker selection was performed by screening 10,000 CpG positions and the final number of selected cell-type markers was set to 500 (as for default).

Consensus clustering was performed following the ConsensusClusterPlus pipeline (maxK = 20, reps = 1000, pItem = 0.8, pFeature = 1 and distance = pearson)[69]. Samples were clustered according to the hierarchical clustering algorithm, ward.D2 method was used for inner linkage and average method was used for the final linkage.

Based on genomic position, the 450 K probes were annotated with the chromatin state, as assigned to normal pancreatic islets (obtained from the integration of ATAC-Seq, DNAme and ChIP-seq data)[26]. Probes associated with the cell-specific transcription factor (TF) checkpoints and associated to the epigenetic states "closed weak enhancer", "lowly-methylated weak enhancer", "open weak enhancer", "closed strong enhancer", "open strong enhancer", "genic enhancer", "insulator" and "polycomb repressed states" were included for looking at the specific TF-checkpoint methylation. Enhancer regions were associated to the nearest gene in the genome using the GenomicRanges R package[70].

Known Imprinting Control Regions (ICRs) were retrieved from WAMIDEX and igc.otago.ac.nz (extended annotations reported in A-GEOD-18809 - Illumina HumanMethylation450 BeadChip (v1.2, extended annotation)).

**Next-generation sequencing**. We sequenced *MEN1*, *DAXX* and *ATRX* genes by semi-conductive sequencing using two Ion Torrent AmpliSeq NGS custom made panels (Life Technologies), one for *ATRX* and one for *MEN1* and *DAXX*, covering whole protein-coding exons. Protein-coding exons were amplified by multiplex polymerase chain reaction using two pools designed by the Ion AmpliSeq Designer and the Ion AmpliSeq Library kit 2.0 according to the manufacturer's recommendations (Life Technologies). Template preparation was performed using the Ion Chef System. Sequencing was performed using the Ion S5. The Torrent Suite 5.10 platform was used for sequence alignment with the hg19 human genome reference. Variant calling was performed with the variant caller and the IonReporter 5.10 software (Life Technologies). The coverage depth was sufficient (quality criteria for a sample to be analysed was minimum 500 reads). The 80 ICGC samples were sequenced by whole-genome sequencing (WGS) within the Scarpa et al. study[47].

**Copy-number aberration analysis**. Genome-wide CNAs were inferred from HM450 signal intensities using the Conumee R package[71]. Chromosome bins and segments were measured (bin size: 50,000 to 5,000,000 bp; minimum number of probes per bin: 15). CNA profile zero-threshold was manually adjusted according to FISH results for the UB-UCL cohort (FISH for chr4 and chr17 was performed for detecting gains, FISH for MEN1 and chr11 was used for detecting losses; see Supplementary Fig. 3a–d). Zero-threshold of the CNA profiles obtained from the ICGC cohort was manually adjusted according to the results obtained from Scarpa et al.[47]. Copy number (CN) of each chromosomal arm for each tumour was obtained calculating the median of the relative copy-number bins. Arm level copy-number data were clustered using Ward's method, Euclidean distance.

**Statistical analysis and graphic representation**. Statistical analysis and graphical representations were performed within the R environment (v. 3.5.0)[72]. Specific packages used in the study and parameters selected are mentioned in the relative method chapters. For disease free survival analysis, the "survival" and "survminer" packages were used[73,74].

**Reporting summary**. Further information on research design is available in the Nature Research Reporting Summary linked to this article.

## Data availability

The authors declare that all data supporting the findings of this study are available within the article and its supplementary data and figures. The datasets generated during the current study (UB-UCL cohort) are available in the ArrayExpress repository (EMBL-EBI, https://www.ebi.ac.uk/arrayexpress/, accession number: E-MTAB-7924). The datasets analysed during the current study (ICGC cohort) are available in the ICGC repository (ICGC, https://icgc.org/, projects: PAEN-AU and PAEN-IT). The datasets of sorted normal hematopoietic cells are available in the GEO repository (Gene Expression Omnibus, http://www.ncbi.nlm.nih.gov/geo/, accession number: GSE35069). The datasets of sorted acinar, duct, alpha and beta pancreatic cells are available in the EGA repository (European Genome-Phenome Archive, https://ega-archive.org/, accession number: EGAS00001002533). The datasets of sorted normal pancreatic fibroblastic cells are available in the GEO repository (Gene Expression Omnibus, http://www.ncbi.nlm.nih.gov/geo/, accession number: GSE80369). Chan et al.[7] dataset is available in the GEO repository (Gene Expression Omnibus, http://www.ncbi.nlm.nih.gov/geo/, accession number: GSE117852).

## Code availability

Details of publicly available software and packages used in the study are given in the "Methods". No custom code or mathematical algorithm that is deemed central to the conclusions was used.

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

## Acknowledgements

We thank Cornelia Schlup and Maja Neuenschwander for technical assistance, Heidi Erika Lisa Tschanz-Lischer, for scientific and bioinformatics support, Tissue Biobank Bern (TBB) for support, Ruth Pidsley and Magali Humbert for helpful discussions. This work was supported by Marie Heim-Vögtlin SNF (PMPDP3_164484) and Tumour Forschung Bern to Ilaria Marinoni, Swiss Cancer League (KLS-4227-08-2017) and Uniscientia Sitzung to Aurel Perren, Cancer Research UK and Experimental Cancer Medicine Centre to Christina Thirlwell and Christodoulos Pipinikas.

## Author contributions

A.D.D. was involved in the study design, was responsible for generation, assembly and analysis of the data and writing of the manuscript. C.P.P. contributed data acquisition and supported in DNA-methylation analysis. M.R.S. gave technical support. K.B., M.D. and A.P. revised pathological data. C.S. provided bioinformatic support. E.V. performed NGS analysis. C.T. contributed acquisition of clinical and pathological data, provided PanNET methylation data from UCL cases and supported in DNA-methylation analysis. C.P.P., K.B., C.S., M.D. and C.T. contributed critical revision of the manuscript for important intellectual content. A.P. and I.M. contributed concept and design of the study, generation, assembly and analysis of the data, writing of the manuscript. C.P.P., C.T, A.P. and I.M. obtained funding.

## Competing interests

The authors declare no competing interests.
