## [Peer Review File · Communications Biology]

Editorial Note: Parts of this Peer Review File have been redacted as indicated to maintain patient confidentiality.

Reviewers' comments:

Reviewer #1 (Remarks to the Author):

The authors studied tumour DNA methylation (DNAm) together with genomic data in order to identify the cell of origin and pathways involved in PanNET progression. They studied genome-wide DNAm data of 125 PanNETs and used sorted α - and β -cells as reference. To confirm cell identity, also ARX and PDX1 expression was studied. Based on epigenetic similarities, PanNETs clustered in α -like, β -like (1/3) and intermediate tumours (2/3).

Interestingly, CNAs were lowest in the former two subgroups. In addition, PanNETs with mutations in DAXX or ATRX were not found, explaining – in part (see below) - the shorter disease-free survival compared to wild type tumours.

The two subgroups (α - and β -like) were low in grading and also primarily T1 and T2, most likely functional (data not shown) and showed – not surprisingly - also a lower relapse rate. In detail, 14/19 of α -like PanNETs were G1 and 14/18 (1 without data) were of low stage (T1 or T2, Fig. 1A and table S1). Similarly, 8/14 β -like PanNETs were G1 and 9/11 (3 without data) were T1 or T2.

DNA methylation corresponded to the cell type assigned by TF expression.

Tumours positive for PDX1 had the lowest risk of relapse. However, DAXX/ATRX status did not improve stratification of ARX+ samples for risk of relapse.

The authors found different mutational spectra, stage of the disease and prognosis and suggest these findings (relevant for 1/3 of patients) as potential means for the prediction of PanNET progression.

In summary, the presented data are novel and clinically relevant. However, some criticisms which should be ruled out

Criticisms:

- The studied patient cohort encompasses a rather large number of small-sized PanNETs thereby limiting the validation parameter DSF/relapse rate, since small Pan NETs are also clinically well-known for slow progression in contrast to larger tumors. Please, explain patient selection.
- Why were patients included lacking correct stage, nodal and metastatic status?
- Furthermore, although a correlation between driver mutation status and epigenetic profiles across all PanNETs was observed, the clinical application in daily practice should be explained. - In the same lines, it is unclear to the reader why the presented epigenetic profiles predict more accurately the risk of relapse, based on the a. m. patient selection.
- Finally, explanation is needed why on the one hand a tumor progression model is presented on the other hand, however, - as stated by the authors - the progression from β -like to intermediate-WT PanNETs, only weakly supported is.

Reviewer #2 (Remarks to the Author):

The overall picture in this report is that DNA methylation/bisulfite sequencing was used in a large primary cohort of pancreatic neuroendocrine tumors (125 PNETs) to categorize, cluster and predict the cell type of origin for PNETs, which the authors infer is an α cell a β cell or a more primitive neuroendocrine progenitor. Their findings are confirmed in a second set of 65 PNETs. The authors suggest, based on DNA methylome analysis, that human PNETs can be divided into three or four broad groups, and these groups correspond to α -like cells, β -like cells, and intermediate cells, and attribute the origin of PNETs as arising from α or β cells in the endocrine pancreas. They further observe that the β -like PNETs display largely benign clinical outcomes, the α cell-like have generally favorable outcomes (with a few exceptions), and the intermediate PNETs generally have unfavorable outcomes, and are associated with additional mutations or CNVs in DAXX, ATRX, and ALT or MEN1, which are well known to be associated with aggressive behavior and poor survival in PNETs. The authors conclude that adding DNA bisulfite sequencing to current targeted mutation sequencing studies in current use clinical use, as well as immunohistochemistry studies (for ARX, PDX1, insulin, glucagon), may provide additional diagnostic and prognostic data for patients with PNETs. Overall, the work is interesting,

but poorly described, and overinterpreted. I am not sure there is much here that is new. Specific comments follow:

Major Comments

1) Perhaps most importantly, I am not sure we have learned much from this study. It is generally accepted that functional insulin- and glucagon-expressing PNETs have a better prognosis than those that produce no hormones. It is also well established that PNETs with DAXX, ATRX, ALT with or without MEN1 mutations are more aggressive, more likely to metastasize, and have worse outcomes than PNETs that lack these mutations. Also, there is no mention of PTEN, mTOR pathways, PI3kinase pathway, TSC1, TSC2 mutations which may also be associated with aggressive tumors.

2) The details and results of the bisulfite DNA sequencing are inadequate and difficult to follow. What exactly was done? Was deep, next-gen bisulfite sequencing performed across the entire genome? Or more likely, chip assays for specific CpGs in specific regions? How were these sites selected? How many CpGs were assessed? Where specifically in the genome were they? And after filtering, 2131 were differentially methylated in PNETs vs. islet cells. This is a tiny number (there are ~30,000 CpG's in the insulin locus at 11p15.5-15.4 alone). And among these, 49 and 51 CpGs were in the regions of the "TFs". Again, these are very tiny numbers. And where in the genome specifically were these 49 and 51 differentially methylated CpGs?? Did they bear any relation to INS, PDX, ARX1, GCG loci or expression? Are they in promoters or enhancers? In imprinting control regions (ICRs)? Related to cell cycle regulatory genes? It is hard to ascribe any significance to such a small number of differentially methylated CpGs.

3) And the "transcription factor" data are confusing. The "TF"s are listed in Table 2. How specifically was this list generated? And to be clear, many of the genes listed do not encode TFs: for example, CDKN1C, BMP5, SMARC1A are not TFs. And what is the point of the "TF" data?

4) The authors interpret the ARX/glucagon and PDX1/insulin expression data as implying that alpha cells and beta cells are the cells of origin in PNETs. I found this unconvincing, and the authors also add a sentence to the Discussion (lines 286-288) acknowledging that time course studies would be required to confirm this hypothesis. But is there evidence that other endocrine progenitor cells in normal islets, such as delta, PP, ghrelin and other neuroendocrine cell types cannot be involved? It is not obvious to this reader that alpha or beta cells must be the cell of origin. Why not a primitive neuroendocrine cell rest left over from fetal pancreas development?

5) It would be important to correlate the findings with a measure of proliferation (Ki67, mitotic index) since these are widely used in tumor prognosis.

6) Line 50-51. "Only a minority of PanNETs are functional, leading to clinical syndromes due to inadequate hormone secretion." This sentence on makes no sense and should be re-written.

Reviewers' comments:

Reviewer #1 (Remarks to the Author):

The authors studied tumour DNA methylation (DNAm) together with genomic data in order to identify the cell of origin and pathways involved in PanNET progression. They studied genome-wide DNAm data of 125 PanNETs and used sorted α - and β -cells as reference. To confirm cell identity, also ARX and PDX1 expression was studied. Based on epigenetic similarities, PanNETs clustered in α -like, β -like (1/3) and intermediate tumours (2/3). Interestingly, CNAs were lowest in the former two subgroups. In addition, PanNETs with mutations in DAXX or ATRX were not found, explaining – in part (see below) - the shorter disease-free survival compared to wild type tumours. The two subgroups (alpha- and beta-like) were low in grading and also primarily T1 and T2, most likely functional (data not shown) and showed – not surprisingly - also a lower relapse rate. In detail, 14/19 of alpha-like PanNETs were G1 and 14/18 (1 without data) were of low stage (T1 or T2, Fig. 1A and table S1). Similarly, 8/14 beta-like PanNETs were G1 and 9/11 (3 without data) were T1 or T2. DNA methylation corresponded to the cell type assigned by TF expression. Tumours positive for PDX1 had the lowest risk of relapse. However, DAXX/ATRX status did not improve stratification of ARX+ samples for risk of relapse. The authors found different mutational spectra, stage of the disease and prognosis and suggest these findings (relevant for 1/3 of patients) as potential means. In summary, the presented data are novel and clinically relevant. However, some criticisms which should be ruled out.

We thank the reviewer for his positive feed-back. The reviewer is correct in stating that alpha-like and beta-like tumours are low in grade and primarily T1 and T2 tumours. Functionality of the tumours is reported in table S1. Notably, only 2/15 (13%, 4 cases NA) of the alpha-like samples were functional (1 glucagonoma and 1 insulinoma), while the vast majority (9/10, 91%, 4 NA) of the beta-like samples are insulinomas.

Criticisms:

1.1: The studied patient cohort encompasses a rather large number of small-sized PanNETs thereby limiting the validation parameter DSF/relapse rate, since small PanNETs are also clinically well-known for slow progression in contrast to larger tumors. Please, explain patient selection.

1.1 We thank the reviewer for this comment. The reviewer is correct, our collective includes 13% (16/123 – 2 NA) of PanNETs smaller than 2cm. This reflects the surgical situation, as our collective is composed of a surgical rather than an oncologic series (Marinoni et al., 2014). We believe that such a collective composition is relevant to predict relapse: while for late stage tumours relapse is very likely to happen, only a portion of small low grade tumours will relapse hence the identification of those patients who need to be closely followed up or who may even need adjuvant therapy, is crucial. Most importantly, since we are interested in identifying cell of origin and way of progression, it is crucial to include in the study small tumours representing early phase of progression. Additionally, the main focus of our research are G1 and G2 tumours which most often have small size, G3 PanNET which always will relapse were excluded.

1.2 Why were patients included lacking correct stage, nodal and metastatic status?

1.2 We agree with the reviewer that we had a relevant number of samples with no information. We were able to retrieve the information for 10 cases, which we added now in table S1. Nevertheless, we decided to include in the clustering analysis also the samples without TNM stage to increase the statistical power (by increasing the number) hence ensuring a more robust molecular tumour classification.

1.3 Furthermore, although a correlation between driver mutation status and epigenetic profiles across all PanNETs was observed, the clinical application in daily practice should be explained.

1.3 We thank you the reviewer for the comment. We recognize that an immediate clinical application is not there yet and additional studies are required. However, our DNA methylation classification, is not simply confirming the clinical relevance of DAXX/ATRX loss, but identifies a new group of samples with poor prognosis (intermediate-WT), which cannot be identified by other types of molecular data except methylation arrays. Moreover, we envision that DNA methylation profiles could be performed in the future on biopsies and even on liquid biopsies. Similar to central nervous system tumours, DNA methylation profile can be used for PanNET classification. Clinical applications could include prognosis and possibly, but not studied here, prediction of therapy response (see discussion lines 316-317).

1.4. In the same lines, it is unclear to the reader why the presented epigenetic profiles predict more accurately the risk of relapse, based on the a. m. patient selection.

1.4. The reviewer is correct, our collective includes only 71/123 (58%, 2 NA) late stage tumours (T3/T4) and 38% (36/95) of the patients present with relapse. Importantly among the patients relapsing, the majority is characterized by DAXX/ATRX mutations (and/or ALT positivity), but an additional part of them are DAXX/ATRX wild type (intermediate-WT) and could only be identified by the study of DNA methylation signatures (see previous point) and Figure 3 A-B-C and Figure S5-C.

1.5 Finally, explanation is needed why on the one hand a tumour progression model is presented on the other hand, however, - as stated by the authors - the progression from β -like to intermediate-WT PanNETs, only weakly supported is.

1.5. We recognize that the reviewer is correct and we modified figure and text accordingly (lines 292-298 and figure 3E): “While CNA, genetic background and epigenetic suggest a progression from alpha cells to alpha-like tumours first and then to intermediate-ADM, we do not have the same evidence supporting an evolution from beta-like tumours to intermediate-WT, even if we cannot completely exclude it. In fact, while a progression from beta-like benign insulinoma to aggressive intermediate-wt tumours is, based on clinical evidences, very unlikely; a direct origin of these tumours from normal adult beta-cells or from a common progenitor is more plausible. Expression of cell type TFs such PDX1 and ARX is equally distributed in intermediate-WT however DNA-methylation profile indicate similarity with beta-cells (Fig. 3D and S6D).”

Reviewer #2 (Remarks to the Author):

The overall picture in this report is that DNA methylation/bisulfite sequencing was used in a large primary cohort of pancreatic neuroendocrine tumors (125 PNETs) to categorize, cluster and predict the cell type of origin for PNETs, which the authors infer is an alpha cell a beta cell or a more primitive neuroendocrine progenitor. Their findings are confirmed in a second set of 65 PNETs. The authors suggest, based on DNA methylome analysis, that human PNETs can be divided into three or four broad groups, and these groups correspond to alpha-like cells, beta-like cells, and intermediate cells, and attribute the origin of PNETs as arising from alpha or beta cells in the endocrine pancreas. They further observe that the beta-like PNETs display largely benign clinical outcomes, the alpha cell-like have generally favorable outcomes (with a few exceptions), and the intermediate PNETs generally have unfavorable outcomes, and are associated with additional mutations or CNVs in DAXX, ATRX, and ALT or MEN1, which are well known to be associated with aggressive behavior and poor survival in PNETs. The authors conclude that adding DNA bisulfite sequencing to current targeted mutation sequencing studies in current use clinical use, as well as immunohistochemistry studies (for ARX, PDX1, insulin, glucagon), may provide additional diagnostic and prognostic data for patients with PNETs. Overall, the work is interesting, but poorly described, and overinterpreted. I am not sure there is much here that is new. Specific comments follow:

We thank you the reviewer for the short overview of our work, however important points are missing: with our approach (not whole genome bisulphite sequencing but HumMeth450 BeadChip array) we were able to define epigenetically novel distinct PanNET subgroups, which overlap with genetic mutations. These subgroups are prognostically relevant, in addition to the mutational status. The subgroups furthermore allow the proposition of a progression model of human PanNET.

Major Comments

2.1 Perhaps most importantly, I am not sure we have learned much from this study. It is generally accepted that functional insulin- and glucagon-expressing PNETs have a better prognosis than those that produce no hormones. It is also well established that PNETs with DAXX, ATRX, ALT with or without MEN1 mutations are more aggressive, more likely to metastasize, and have worse outcomes than PNETs that lack these mutations. Also, there is no mention of PTEN, mTOR pathways, PI3kinase pathway, TSC1, TSC2 mutations which may also be associated with aggressive tumors.

2.1 In our work, we clearly described for the first time on a large number of samples 4 epigenetic subgroups, with distinct: epigenetic, genetic, genomic aberration and prognosis. Our DNA methylation classification, besides confirming the clinical relevance of DAXX/ATRX loss, also identifies a new group of samples with poor prognosis (intermediate-WT), which cannot be detected by other types of molecular data.

The scope of the study goes beyond the mere description of prognosis associated with DAXX/ATRX mutation or functional tumours. It further focuses on identifying specific cell of origin and way of progression using DNA methylation alongside with gene mutations

and CNA and on using these parameters to group clinically distinct PanNETs. We have been able to describe two distinct cells of origin for PanNETs, based on DNA-methylation, confirming the recent findings observed in Cejas et al. (Cejas et al., 2019), who however performed the study on a very limited number of samples (n=21). Additionally, Cejas and co-workers used a super-enhancer signature, while our approach, looking at genome-wide methylation signatures, is much broader. In conclusion, our findings move one-step forward in better defining PanNET groups, cell of origin and clinical implication.

The reviewer is correct in saying that other mutations such as member of mTOR pathways may be associated with aggressive behaviour. Following on the reviewer comment we added the information in table S1. These mTOR pathway mutations are usually occurring in about 15% of sporadic PanNETs (Jiao et al., 2011; Scarpa et al., 2017) and in our cluster analysis show a slight enrichment in the intermediate-ADM group (14/80 (17,5%) samples in the ICGC cohort show mutations in the mTOR pathway - 11/46 (24%) of the intermediate-ADM samples, 2/12 (17%) of the intermediate-WT samples, 1/15 (6%) of the alpha-like samples, 0/7 of the beta-like samples).

2.2. The details and results of the bisulfite DNA sequencing are inadequate and difficult to follow. What exactly was done? Was deep, next-gen bisulfite sequencing performed across the entire genome? Or more likely, chip assays for specific CpGs in specific regions? How were these sites selected? How many CpGs were assessed? Where specifically in the genome were they? And after filtering, 2131 were differentially methylated in PNETs vs. islet cells. This is a tiny number (there are ~30,000 CpG's in the insulin locus at 11p15.5-15.4 alone). And among these, 49 and 51 CpGs were in the regions of the "TFs". Again, these are very tiny numbers. And where in the genome specifically were these 49 and 51 differentially methylated CpGs?? Did they bear any relation to INS, PDX, ARX1, GCG loci or expression? Are they in promoters or enhancers? In imprinting control regions (ICRs)? Related to cell cycle regulatory genes? It is hard to ascribe any significance to such a small number of differentially methylated CpGs.

2.2 We thought that the method and technical details were exhaustively explained in the material and methods section, nevertheless upon the reviewer comment we made some repetitions in the results section. As reported in the material and methods we did not perform bisulfite sequencing but we used instead an array based approach. To follow a point to point answer to the single questions of comment number 2.2

2.2.1 The details and results of the bisulfite DNA sequencing are inadequate and difficult to follow. What exactly was done? Was deep, next-gen bisulfite sequencing performed across the entire genome? Or more likely, chip assays for specific CpGs in specific regions

2.2.1 As stated in the materials and methods section (line 368) we did not perform bisulfite DNA sequencing but we instead performed a chip assay, in detail the HumMeth450 BeadChip (Illumina HM450) was used to generate the data. To make it clearer we repeated it in the results section (line 84).

2.2.2 How were these sites selected? How many CpGs were assessed? Where specifically in the genome were they?

2.2.2 *The HumMeth450 BeadChip includes 485,512 probes. The assay includes sites spread in the whole genome with enrichment for CpG islands and promoter regions (Sandoval et al., 2011). As stated in the materials and methods section (lines 370-372), probe filtering was performed as described in the ChAMP pipeline (Morris et al., 2014) and in details we filtered for: SNPs (SNP list generated from Zhou's Nucleic Acids Research paper in 2017 (Zhou et al., 2017)), low quality probes (detection p value >0.01 and/or probes with <3 beads in at least 5% of samples per probe), all non-CpG probes, multi-hit probes (list come from Nordlund's Genome Biology paper in 2013 (Nordlund et al., 2013)) and finally for probes located on chromosome X and Y. After filtration, the PanNET sample matrix retained 363,665 probes and 125 samples (added at lines 88-89). All the relevant publications are referenced in the paper.*

2.2.3 **And after filtering, 2131 were differentially methylated in PNETs vs. islet cells. This is a tiny number (there are ~30,000 CpG's in the insulin locus at 11p15.5-1 5.4 alone).**

2.2.3 *As already reported above, in our analysis we used chip based methylation data rather than bisulfite DNA sequencing data. The 2,131 differentially methylated sites refer to the comparison between samples of sorted alpha (n=2 samples) and sorted beta (n=2 samples) normal cells. Only sites with high differences in methylation (>20%) and low BH adjusted p value (adj. p value < 0.001) were selected in this case. The strict statistical thresholds and the low number of replicates might explain the number of CpGs reported. Nevertheless these numbers comply with numbers found in other studies, more in detail, Neiman and colleagues, comparing alpha and beta cell methylation (using the HumMeth450 BeadChip) show that: "genes active in one cell type and silent in the other tend to share demethylated promoters, while methylation differences between α - and 13-cells are concentrated in enhancers". Additionally, "Among over 450,000 sites analyzed from all autosomal chromosomes, they found 745 sites uniquely hypomethylated in 13-cells, 353 sites uniquely hypomethylated in α -cells, and 3,753 sites commonly hypomethylated in both cell types" (Neiman et al., 2017). Table S2 includes all the 2,131 selected sites. In the table is reported location and features of the sites, we have added here a column reporting the specific chromatin state detected in normal islets by M. Thurner and colleagues (Thurner et al., 2018) as well as annotation about the imprinting control regions (only one probe is located on a paternal ICR). The attached figures depict number of DMPs (red) and total number of probes of the 450K array in the 15 pancreatic islet specific chromatin states (A) and the percentage of DMPs relative to the total number of probes in the specific chromatin states (B).*

2.2.4 And among these, 49 and 51 CpGs were in the regions of the “TFs”. Again, these are very tiny numbers. And where in the genome specifically were these 49 and 51 differentially methylated CpGs?? Did they bear any relation to INS, PDX, ARX1, GCG loci or expression? Are they in promoters or enhancers? In imprinting control regions (ICRs)? Related to cell cycle regulatory genes? It is hard to ascribe any significance to such a small number of differentially methylated CpGs.

2.2.4 As stated above we performed the analysis by using the HumMeth450 BeadChip, indeed this assay shows underrepresentation of enhancer/regulatory regions (Sandoval et al., 2011) and we agree with the reviewer that the selection of CpG sites located at regulatory regions of the genome (“closed weak enhancer”, “lowly-methylated weak enhancer”, “open weak enhancer”, “closed strong enhancer”, “open strong enhancer”, “genic enhancer”, “insulator” and “polycomb repressed states” – described in materials and methods, lines 392-398) results in a tiny number of probes (in total 102,666 probes).

The selection of sites, respectively to specific genes, was based on the list generated by Muraro et al. (Muraro et al., 2016) (line 108). In the study they describe the top 10 TF checkpoints specific for each of the 5 pancreatic endocrine cell types.

The total number of probes (irrespective from genomic location and located on autosomal chromosomes) associated to the alpha/beta TF checkpoints selected, account for respectively 183 and 230 sites (alpha genes included: FEV, RFX6, MAFB, PGR, IRX2, PTGER3, LDB2; beta genes included: SIX3, MNX1, PDX1, CDKN1C, SMAD9, TFCP2L1, MAFA, SIX2, BMP5 – listed in table 2). Considering that about $\frac{1}{4}$ of the total probes is belonging to regulatory regions the proportions are retained.

A better description of probe selection is included in lines 111-114.

Despite the small number of probes selected (49 and 51) we see a clear correlation between the clusters of samples defined by cell type specific TF checkpoints regulatory regions DNA methylation and expression of PDX1 and ARX as well as with the clusters defined by the phyloepigenetic trees (see figure 1 and table S1). [Redacted] we see high expression of the selected alpha/beta TF checkpoints in the alpha-like and

beta-like tumours, respectively and gradual downregulation in the intermediate samples (see heatmap below). [Redacted]

[Redacted]

The analysis we have performed is independent from any differential methylation between samples, providing a different approach for confirming the segregation of the samples according to the relative hypothetical cell of origin, when specific regions are selected. We have included as part of the manuscript 2 tables listing the 49 and 51 probes which are located in regulatory regions of the genome closed to specific alpha and beta TF checkpoints (table S3 and S4). None of the selected probes is located on an ICR (known ICRs were retrieved from WAMIDEX and igc.otago.ac.nz – extended annotations reported in A-GEOD-18809 - Illumina HumanMethylation450 BeadChip (v1.2, extended annotation) – description added at lines 399-401).

2.3 And the “transcription factor” data are confusing. The “TF”s are listed in Table 2. How specifically was this list generated? And to be clear, many of the genes listed do not encode TFs: for example, CDKN1C, BMP5, SMARCA1 are not TFs. And what is the point of the “TF” data?

2.3 As referenced, the list reported in table 2 was generated in the study of Muraro et al., the selection refers to figure 1E (Muraro et al., 2016). In this study the authors “... selected those genes that have been reported to function as transcription factors using the TFcheckpoint database (Chawla et al., 2013) (Table S4). Several genes and transcription factors found here have never been reported as markers for specific cell types of the human pancreas...”

We recognized that we have oversimplified the definition of the genes as transcription factors, and we have corrected the text accordingly (TF checkpoints). Indeed some of these proteins are not direct transcription factors but involved in the regulation of transcription (CDKN1C reducing phosphorylation of the RNA polymerase II (pol II) C-terminal domain (CTD) (Ma et al., 2010), BMP5 acts as coactivator of the SMAD4 pathway and inhibitor of the STAT/JAK pathway (Chen et al., 2018), and SMARCA1 acting as ATPase which contributes to the chromatin remodeling complex involved in transcription (Peterson and Workman, 2000)). Additionally, it has been shown that CDKN1C is upregulated in NEUROG3 expressing cells, blocking cell cycle of the progenitor cells (Georgia et al., 2006). Similarly, induction of BMP signalling modulates cell fate choice between liver and pancreas (Rodríguez-Seguel et al., 2013). SMARCA1 shows downregulation with aging in islet cells (Rankin and Kushner, 2010) and specific developmental roles (Lazzaro and Picketts, 2001). Given the implication in regulating transcription processes and the involvement into islet cell development we decided to include the listed genes.

As pointed above the analysis we have performed is independent from any differential methylation between samples, providing a different approach for confirming the segregation of the samples according to the relative hypothetical cell of origin, when specific regions are selected. Furthermore, this analysis shows the similarity of intermediate-ADM to alpha-like tumours and the most likely different nature of intermediate-WT tumours, as discussed in lines 234-236.

2.4 The authors interpret the ARX/glucagon and PDX1/insulin expression data as implying that alpha cells and beta cells are the cells of origin in PNETs. I found this unconvincing, and the authors also add a sentence to the Discussion (lines 286-288) acknowledging that time course studies would be required to confirm this hypothesis. But is there evidence that other endocrine progenitor cells in normal islets, such as delta, PP, ghrelin and other neuroendocrine cell types cannot be involved? It is not obvious to this reader that alpha or beta cells must be the cell of origin. Why not a primitive neuroendocrine cell rest left over from fetal pancreas development?

2.4 We would agree with the reviewer if the data on ARX/glucagon and PDX1/insulin expression would stand alone. We recognize that we might have not been clear enough in the text in explaining our hypothesis. ARX and PDX1 expression have been used only to confirm the cell type identified by DNA methylation analysis and per se they are not an indicator of a specific cell of origin. On the contrary, our hypothesis on the cell of origin is based on the DNA methylation profiles and not on ARX/PDX1 expression.

Indeed, comparison of PanNET DNA methylation with the ones of sorted pancreatic cells including alpha, beta, acinar, ductal cells showed that a group of PanNET clearly resemble alpha and beta endocrine cells (alpha- and beta-like tumours), while an intermediate group shows low degrees of similarity to alpha or any cell type (see figure 1A, S2B and table S1). Nevertheless intermediate-ADM are more similar to alpha cells than other cell types (see figure 1A and table S1). Based on these evidences and given the recent publication in Nature Medicine from Cejas and colleagues (Cejas et al., 2019), we concluded that subgroups of tumours may originate either from alpha or from beta cells.

As stated in the introduction, DNA methylation profile is highly conserved along differentiation. For example, it was shown that cells with a common embryonic origin share more overlap in their DNA methylation than cells from the same tissues but with different origin (Kundaje et al., 2015), suggesting that common developmental origin is a primary determinant for global DNA methylation. For these reasons, epigenome is suitable to learn biologically meaningful relationships between cell type, tissues and lineages. As mentioned in the text, this approach has been used already in several studies in brain tumours to identify distinct cell of origin and developmental path (Gibson et al., 2010; Northcott et al., 2017) which were confirmed by scRNA-seq experiments (Hovestadt et al., 2019). Similarly, integrative analysis of colorectal adenoma, carcinoma, and crypt section methylomes shows that the methylation patterns of specific intestinal cell differentiation stages are maintained during colorectal carcinogenesis (Bormann et al., 2018).

In conclusion, we agree with the reviewer that intermediate tumours might arise from an adult ϵ , δ or PP cells, or even from a common progenitor cell but there is no methylome data or other evidence that supports this hypothesis. DNA methylation of alpha and beta like tumours instead strongly speaks in favour of an origin from alpha and beta endocrine cells. We do not exclude that intermediate tumours may originate from a precursor cell, especially the intermediate-WT for which DNA methylation profile do not clearly indicate a specific cell similarity. Upon the reviewer comments we rephrase this part in the discussion (lines 292-298).

2.5 It would be important to correlate the findings with a measure of proliferation (Ki67, mitotic index) since these are widely used in tumor prognosis.

2.5 We appreciate the reviewer suggestion and have performed the analysis for a subset of 56 samples for which data on Ki-67 was available. Results are shown in figure S6. The results are described in line 179 and Ki-67 values (%) are reported in table S1.

2.6 Line 50-51. “Only a minority of PanNETs are functional, leading to clinical syndromes due to inadequate hormone secretion.” This sentence on makes no sense and should be re-written.

2.6 This sentence has been corrected (lines 50-51): “Only a minority of PanNETs are functional, secreting inadequate hormones that lead to clinical syndromes”.

References

- Bormann, F., Rodríguez-Paredes, M., Lasitschka, F., Edelmann, D., Musch, T., Benner, A., Bergman, Y., Dieter, S.M., Ball, C.R., Glimm, H., Linhart, H.G., Lyko, F., 2018. Cell-of-Origin DNA Methylation Signatures Are Maintained during Colorectal Carcinogenesis. *Cell Reports* 23, 3407–3418. <https://doi.org/10.1016/j.celrep.2018.05.045>
- Cejas, P., Drier, Y., Dreijerink, K.M.A., Brosens, L.A.A., Deshpande, V., Epstein, C.B., Conemans, E.B., Morsink, F.H.M., Graham, M.K., Valk, G.D., Vriens, M.R., Castillo, C.F.-D., Ferrone, C.R., Adar, T., Bowden, M., Whitton, H.J., Da Silva, A., Font-Tello, A., Long, H.W., Gaskell, E., Shoresh, N., Heaphy, C.M., Sicinska, E., Kulke, M.H., Chung, D.C., Bernstein, B.E., Shivdasani, R.A., 2019. Enhancer signatures stratify and predict outcomes of non-functional pancreatic neuroendocrine tumors. *Nat. Med.* 25, 1260–1265. <https://doi.org/10.1038/s41591-019-0493-4>
- Chan, C.S., Laddha, S.V., Lewis, P.W., Koletsky, M.S., Robzyk, K., Da Silva, E., Torres, P.J., Untch, B.R., Li, J., Bose, P., Chan, T.A., Klimstra, D.S., Allis, C.D., Tang, L.H., 2018. ATRX, DAXX or MEN1 mutant pancreatic neuroendocrine tumors are a distinct alpha-cell signature subgroup. *Nat Commun* 9, 4158. <https://doi.org/10.1038/s41467-018-06498-2>
- Chawla, K., Tripathi, S., Thommesen, L., Lægreid, A., Kuiper, M., 2013. TFcheckpoint: a curated compendium of specific DNA-binding RNA polymerase II transcription factors. *Bioinformatics* 29, 2519–2520. <https://doi.org/10.1093/bioinformatics/btt432>
- Chen, E., Yang, F., He, H., Li, Q., Zhang, W., Xing, J., Zhu, Z., Jiang, J., Wang, H., Zhao, X., Liu, R., Lei, L., Dong, J., Pei, Y., Yang, Y., Pan, J., Zhang, P., Liu, S., Du, L., Zeng, Y., Yang, J., 2018. Alteration of tumor suppressor BMP5 in sporadic colorectal cancer: a genomic and transcriptomic profiling based study. *Molecular Cancer* 17, 176. <https://doi.org/10.1186/s12943-018-0925-7>
- Georgia, S., Soliz, R., Li, M., Zhang, P., Bhushan, A., 2006. p57 and Hes1 coordinate cell cycle exit with self-renewal of pancreatic progenitors. *Developmental Biology* 298, 22–31. <https://doi.org/10.1016/j.ydbio.2006.05.036>
- Gibson, P., Tong, Y., Robinson, G., Thompson, M.C., Currell, D.S., Eden, C., Kranenburg, T.A., Hogg, T., Poppleton, H., Martin, J., Finkelstein, D., Pounds, S., Weiss, A., Patay, Z., Scoggins, M., Ogg, R., Pei, Y., Yang, Z.-J., Brun, S., Lee, Y., Zindy, F., Lindsey, J.C., Taketo, M.M., Boop, F.A., Sanford, R.A., Gajjar, A., Clifford, S.C., Roussel, M.F., McKinnon, P.J., Gutmann, D.H., Ellison, D.W., Wechsler-Reya, R., Gilbertson, R.J., 2010. Subtypes of medulloblastoma have distinct developmental origins. *Nature* 468, 1095–1099. <https://doi.org/10.1038/nature09587>
- Hovestadt, V., Smith, K.S., Bihannic, L., Filbin, M.G., Shaw, M.L., Baumgartner, A., DeWitt, J.C., Groves, A., Mayr, L., Weisman, H.R., Richman, A.R., Shore, M.E., Goumnerova, L., Rosencrance, C., Carter, R.A., Phoenix, T.N., Hadley, J.L., Tong, Y., Houston, J., Ashmun, R.A., DeCuyper, M., Sharma, T., Flasch, D., Silkov, A., Ligon, K.L., Pomeroy, S.L., Rivera, M.N., Rozenblatt-Rosen, O., Ruser, J.M., Wechsler-Reya, R.J., Li, X.-N., Peyrl, A., Gojo, J., Kirchhofer, D., Lötsch, D., Czech, T., Dorfer, C., Haberler, C., Geyeregger, R., Halfmann, A., Gawad, C., Easton, J., Pfister, S.M., Regev, A., Gajjar, A., Orr, B.A., Slavc, I., Robinson, G.W., Bernstein, B.E., Suvà, M.L., Northcott, P.A., 2019. Resolving

- medulloblastoma cellular architecture by single-cell genomics. *Nature* 572, 74–79. <https://doi.org/10.1038/s41586-019-1434-6>
- Jiao, Y., Shi, C., Edil, B.H., de Wilde, R.F., Klimstra, D.S., Maitra, A., Schulick, R.D., Tang, L.H., Wolfgang, C.L., Choti, M.A., Velculescu, V.E., Diaz, L.A., Vogelstein, B., Kinzler, K.W., Hruban, R.H., Papadopoulos, N., 2011. DAXX/ATRX, MEN1, and mTOR pathway genes are frequently altered in pancreatic neuroendocrine tumors. *Science* 331, 1199–1203. <https://doi.org/10.1126/science.1200609>
- Kundaje, A., Meuleman, W., Ernst, J., Bilenky, M., Yen, A., Heravi-Moussavi, A., Kheradpour, P., Zhang, Z., Wang, J., Ziller, M.J., Amin, V., Whitaker, J.W., Schultz, M.D., Ward, L.D., Sarkar, A., Quon, G., Sandstrom, R.S., Eaton, M.L., Wu, Y.-C., Pfenning, A.R., Wang, X., Claussnitzer, M., Liu, Y., Coarfa, C., Harris, R.A., Shores, N., Epstein, C.B., Gjoneska, E., Leung, D., Xie, W., Hawkins, R.D., Lister, R., Hong, C., Gascard, P., Mungall, A.J., Moore, R., Chuah, E., Tam, A., Canfield, T.K., Hansen, R.S., Kaul, R., Sabo, P.J., Bansal, M.S., Carles, A., Dixon, J.R., Farh, K.-H., Feizi, S., Karlic, R., Kim, A.-R., Kulkarni, A., Li, D., Lowdon, R., Elliott, G., Mercer, T.R., Neph, S.J., Onuchic, V., Polak, P., Rajagopal, N., Ray, P., Sallari, R.C., Siebenthal, K.T., Sinnott-Armstrong, N.A., Stevens, M., Thurman, R.E., Wu, J., Zhang, B., Zhou, X., Beaudet, A.E., Boyer, L.A., Jager, P.L.D., Farnham, P.J., Fisher, S.J., Haussler, D., Jones, S.J.M., Li, W., Marra, M.A., McManus, M.T., Sunyaev, S., Thomson, J.A., Tlsty, T.D., Tsai, L.-H., Wang, W., Waterland, R.A., Zhang, M.Q., Chadwick, L.H., Bernstein, B.E., Costello, J.F., Ecker, J.R., Hirst, M., Meissner, A., Milosavljevic, A., Ren, B., Stamatoyannopoulos, J.A., Wang, T., Kellis, M., 2015. Integrative analysis of 111 reference human epigenomes. *Nature* 518, 317–330. <https://doi.org/10.1038/nature14248>
- Lazzaro, M.A., Picketts, D.J., 2001. Cloning and characterization of the murine Imitation Switch (ISWI) genes: differential expression patterns suggest distinct developmental roles for Snf2h and Snf2l. *Journal of Neurochemistry* 77, 1145–1156. <https://doi.org/10.1046/j.1471-4159.2001.00324.x>
- Ma, Y., Chen, L., Wright, G.M., Pillai, S.R., Chellappan, S.P., Cress, W.D., 2010. CDKN1C Negatively Regulates RNA Polymerase II C-terminal Domain Phosphorylation in an E2F1-dependent Manner. *J. Biol. Chem.* 285, 9813–9822. <https://doi.org/10.1074/jbc.M109.091496>
- Marinoni, I., Kurrer, A.S., Vassella, E., Dettmer, M., Rudolph, T., Banz, V., Hunger, F., Pasquinelli, S., Speel, E.-J., Perren, A., 2014. Loss of DAXX and ATRX are associated with chromosome instability and reduced survival of patients with pancreatic neuroendocrine tumors. *Gastroenterology* 146, 453-460.e5. <https://doi.org/10.1053/j.gastro.2013.10.020>
- Morris, T.J., Butcher, L.M., Feber, A., Teschendorff, A.E., Chakravarthy, A.R., Wojdacz, T.K., Beck, S., 2014. ChAMP: 450k Chip Analysis Methylation Pipeline. *Bioinformatics* 30, 428–430. <https://doi.org/10.1093/bioinformatics/btt684>
- Muraro, M.J., Dharmadhikari, G., Grün, D., Groen, N., Dielen, T., Jansen, E., van Gorp, L., Engelse, M.A., Carlotti, F., de Koning, E.J.P., van Oudenaarden, A., 2016. A Single-Cell Transcriptome Atlas of the Human Pancreas. *Cell Systems* 3, 385-394.e3. <https://doi.org/10.1016/j.cels.2016.09.002>
- Neiman, D., Moss, J., Hecht, M., Magenheimer, J., Piyanzin, S., Shapiro, A.M.J., Koning, E.J.P. de, Razin, A., Cedar, H., Shemer, R., Dor, Y., 2017. Islet cells

- share promoter hypomethylation independently of expression, but exhibit cell-type-specific methylation in enhancers. *PNAS* 114, 13525–13530.
<https://doi.org/10.1073/pnas.1713736114>
- Nordlund, J., Bäcklin, C.L., Wahlberg, P., Busche, S., Berglund, E.C., Eloranta, M.-L., Flaegstad, T., Forestier, E., Frost, B.-M., Harila-Saari, A., Heyman, M., Jónsson, Ó.G., Larsson, R., Palle, J., Rönnblom, L., Schmiegelow, K., Sinnott, D., Söderhäll, S., Pastinen, T., Gustafsson, M.G., Lönnerholm, G., Syvänen, A.-C., 2013. Genome-wide signatures of differential DNA methylation in pediatric acute lymphoblastic leukemia. *Genome Biology* 14, r105.
<https://doi.org/10.1186/gb-2013-14-9-r105>
- Northcott, P.A., Buchhalter, I., Morrissy, A.S., Hovestadt, V., Weischenfeldt, J., Ehrenberger, T., Gröbner, S., Segura-Wang, M., Zichner, T., Rudneva, V.A., Warnatz, H.-J., Sidiropoulos, N., Phillips, A.H., Schumacher, S., Kleinheinz, K., Waszak, S.M., Erkek, S., Jones, D.T.W., Worst, B.C., Kool, M., Zapatka, M., Jäger, N., Chavez, L., Hutter, B., Bieg, M., Paramasivam, N., Heinold, M., Gu, Z., Ishaque, N., Jäger-Schmidt, C., Imbusch, C.D., Jugold, A., Hübschmann, D., Risch, T., Amstislavskiy, V., Gonzalez, F.G.R., Weber, U.D., Wolf, S., Robinson, G.W., Zhou, X., Wu, G., Finkelstein, D., Liu, Y., Cavalli, F.M.G., Luu, B., Ramaswamy, V., Wu, X., Koster, J., Ryzhova, M., Cho, Y.-J., Pomeroy, S.L., Herold-Mende, C., Schuhmann, M., Ebinger, M., Liau, L.M., Mora, J., McLendon, R.E., Jabado, N., Kumabe, T., Chuah, E., Ma, Y., Moore, R.A., Mungall, A.J., Mungall, K.L., Thiessen, N., Tse, K., Wong, T., Jones, S.J.M., Witt, O., Milde, T., Von Deimling, A., Capper, D., Korshunov, A., Yaspo, M.-L., Kriwacki, R., Gajjar, A., Zhang, J., Beroukhi, R., Fraenkel, E., Korbel, J.O., Brors, B., Schlesner, M., Eils, R., Marra, M.A., Pfister, S.M., Taylor, M.D., Lichter, P., 2017. The whole-genome landscape of medulloblastoma subtypes. *Nature* 547, 311–317. <https://doi.org/10.1038/nature22973>
- Peterson, C.L., Workman, J.L., 2000. Promoter targeting and chromatin remodeling by the SWI/SNF complex. *Current Opinion in Genetics & Development* 10, 187–192.
[https://doi.org/10.1016/S0959-437X\(00\)00068-X](https://doi.org/10.1016/S0959-437X(00)00068-X)
- Rankin, M.M., Kushner, J.A., 2010. Aging induces a distinct gene expression program in mouse islets. *Islets* 2, 345–352. <https://doi.org/10.4161/isl.2.6.13376>
- Rodríguez-Seguel, E., Mah, N., Naumann, H., Pongrac, I.M., Cerdá-Esteban, N., Fontaine, J.-F., Wang, Y., Chen, W., Andrade-Navarro, M.A., Spagnoli, F.M., 2013. Mutually exclusive signaling signatures define the hepatic and pancreatic progenitor cell lineage divergence. *Genes Dev.* 27, 1932–1946.
<https://doi.org/10.1101/gad.220244.113>
- Sandoval, J., Heyn, H., Moran, S., Serra-Musach, J., Pujana, M.A., Bibikova, M., Esteller, M., 2011. Validation of a DNA methylation microarray for 450,000 CpG sites in the human genome. *Epigenetics* 6, 692–702.
<https://doi.org/10.4161/epi.6.6.16196>
- Scarpa, A., Chang, D.K., Nones, K., Corbo, V., Patch, A.-M., Bailey, P., Lawlor, R.T., Johns, A.L., Miller, D.K., Mafficini, A., Rusev, B., Scardoni, M., Antonello, D., Barbi, S., Sikora, K.O., Cingarlini, S., Vicentini, C., McKay, S., Quinn, M.C.J., Bruxner, T.J.C., Christ, A.N., Harliwong, I., Idrisoglu, S., McLean, S., Nourse, C., Nourbakhsh, E., Wilson, P.J., Anderson, M.J., Fink, J.L., Newell, F., Waddell, Nick, Holmes, O., Kazakoff, S.H., Leonard, C., Wood, S., Xu, Q., Nagaraj, S.H., Amato, E., Dalai, I., Bersani, S., Cataldo, I., Dei Tos, A.P., Capelli, P., Davi, M.V., Landoni, L., Malpaga, A., Miotto, M., Whitehall, V.L.J.,

- Leggett, B.A., Harris, J.L., Harris, J., Jones, M.D., Humphris, J., Chantrill, L.A., Chin, V., Nagrial, A.M., Pajic, M., Scarlett, C.J., Pinho, A., Rooman, I., Toon, C., Wu, J., Pinese, M., Cowley, M., Barbour, A., Mawson, A., Humphrey, E.S., Colvin, E.K., Chou, A., Lovell, J.A., Jamieson, N.B., Duthie, F., Gingras, M.-C., Fisher, W.E., Dagg, R.A., Lau, L.M.S., Lee, M., Pickett, H.A., Reddel, R.R., Samra, J.S., Kench, J.G., Merrett, N.D., Epari, K., Nguyen, N.Q., Zeps, N., Falconi, M., Simbolo, M., Butturini, G., Van Buren, G., Partelli, S., Fassan, M., Australian Pancreatic Cancer Genome Initiative, Khanna, K.K., Gill, A.J., Wheeler, D.A., Gibbs, R.A., Musgrove, E.A., Bassi, C., Tortora, G., Pederzoli, P., Pearson, J.V., Waddell, Nicola, Biankin, A.V., Grimmond, S.M., 2017. Whole-genome landscape of pancreatic neuroendocrine tumours. *Nature* 543, 65–71. <https://doi.org/10.1038/nature21063>
- Thurner, M., van de Bunt, M., Torres, J.M., Mahajan, A., Nylander, V., Bennett, A.J., Gaulton, K.J., Barrett, A., Burrows, C., Bell, C.G., Lowe, R., Beck, S., Rakyan, V.K., Gloyn, A.L., McCarthy, M.I., 2018. Integration of human pancreatic islet genomic data refines regulatory mechanisms at Type 2 Diabetes susceptibility loci. *eLife* 7, e31977. <https://doi.org/10.7554/eLife.31977>
- Zhou, W., Laird, P.W., Shen, H., 2017. Comprehensive characterization, annotation and innovative use of Infinium DNA methylation BeadChip probes. *Nucleic Acids Res.* 45, e22. <https://doi.org/10.1093/nar/gkw967>

Reviewer #1 (Remarks to the Author):

The authors have carefully answered most of the critical questions raised by the reviewers. Therefore, I see no objection to accept the revised manuscript in the present form

Reviewer #3 (Remarks to the Author):

In Di Domenico et al., the authors use the Illumina 450K platform to determine the methylation patterns of 45 PanNets. These data were then combined with similar results from 80 previously assessed PanNets. Using published 450K data from 2 samples each of pancreatic alpha-cells and beta-cells, the authors identified 2131 differentially methylated loci (DMLs) between the two islet cell types. These DMLs are then used to cluster the 125 PanNets into 3 groups, a small group with a methylation profile very similar to alpha-cells, another small group similar to beta-cells, and the majority of tumors as "intermediate" which had weak methylation pattern similarity to alpha-cells, but very little similarity to beta-cells. Mutation analysis showed that most beta-like PanNets did not have mutations in *MEN1*, *DAXX*, or *ATRX*, genes frequently mutated in PanNets. In contrast, roughly half of the alpha-like tumors had mutations in *MEN1*, while the other half were wild-type for *MEN1*, *DAXX*, or *ATRX*. The majority of intermediate PanNets had mutations in *MEN1* and/or *DAXX/ATRX*. Tumors were also analyzed for copy number aberrations; both alpha-like and beta-like tumors had very few CNAs, whereas intermediate tumors had many.

The staining patterns of 61 of these PanNets for transcription factors ARX and PDX1, was consistent with the methylation-based similarity. These results were confirmed in an independent cohort of 32 PanNets. Comparison of the epigenetic differentiation status was found to be associated with clinical outcome with most alpha-like and beta-like tumors being of low grade (G1) and low stage (T1 or T2). In contrast, intermediate tumors were generally of higher grade and stage, contained an increase number of proliferative cells, and a higher rate of relapse. Intermediate tumors also had a lower disease-free survival than alpha-like or beta-like tumors. The authors conclude that tumor methylation could predict the risk of relapse more accurately than IHC for PDX1, ARX1, or DAXX/ATRX .

The authors suggest that PanNETs arise along 2 possible pathways, one originating from alpha-cells and the other from beta-cells. Although these results might be expected based on previous clinical and histological data, the results presented in this study provide molecular evidence and further characterizes the majority of PanNETS, so-called intermediate PanNETS into two subgroups. Furthermore, the results confirm that the less differentiated PanNETS are the most likely to metastasize and/or lead to relapses.

This study provides new information as to the origin and clinical progression of the different types of PanNETS, which may ultimately inform diagnosis and treatment strategies. It will be interesting to compare the results presented here with similar studies that classify PanNETS by other molecular approaches.

This manuscript is fairly well written but could be improved with respect to English grammar by an editor. The study is largely bioinformatics analyses, and in some cases, the analyses could be more clearly explained for nonexperts in bioinformatics.

Minor points:

1. For those not familiar with PanNETs, clarify if DAXX/ATRX means a mutation in either or both gene.
2. Line 56, clarify "characterized by 'stemness' transcripts"
3. Lines 81-87. Make it more clear, what 450K assays were carried out by the current authors and which are from other sources.

4. Line 88, explain meaning of "tumor DNA matrix"
5. Line 93, briefly explain how "cell-type deconvolution" was done.
6. Line 96, "phyloepigenies that" would be better as "phyloepigenies of tumors that"..
7. Line 108, explain what TF checkpoints are.
8. Line 109, should "regulatory states" be "regulatory chromatin states"?
9. Line 113, what does "More in detail" mean, Specifically?
10. Line 125, should "enriched for MEN only mutations" be better as " enriched for tumors with only MEN1 mutations"?
11. Line 173 "and intermediate tumors" (from cohort 1).
12. Line 210, should "peculiar" actually ne "particular"